# The GPU version of LICOM3 under the HIP framework and its large-scale application

Pengfei Wang[1,3], Jinrong Jiang[2, 4*], Pengfei Lin[1,4*], Mengrong Ding[1], Junlin Wei[2],Feng Zhang[2],Lian Zhao[2], Yiwen Li[1], Zipeng Yu[1], Weipeng Zheng[1,4], Yongqiang Yu[1,4], Xuebin Chi[2, 4] and Hailong Liu[1,4*]

[1]State Key Laboratory of Numerical Modeling for Atmospheric Sciences and Geophysical Fluid Dynamics (LASG), Institute of Atmospheric Physics (IAP), Chinese Academy of Sciences (CAS), Beijing 100029, China
[2]Computer Network Information Center, Chinese Academy of Sciences, Beijing 100190, China
[3]Center for Monsoon System Research (CMSR), Institute of Atmospheric Physics, Chinese Academy of Sciences, Beijing 100190, China
[4]University of Chinese Academy of Sciences, Beijing 100049, China

*Correspondence*: Drs. Jinrong Jiang [jjr@sccas.cn], Pengfei Lin [linpf@mail.iap.ac.cn] and Hailong Liu [lhl@lasg.iap.ac.cn]

**Abstract.** A high-resolution (1/20°) global ocean general circulation model with graphics processing unit (GPU) code implementations is developed based on the LASG/IAP Climate System Ocean Model version 3 (LICOM3) under a heterogeneous-compute interface for portability (HIP) framework. The dynamic core and physics package of LICOM3 are both ported to the GPU, and 3-dimensional parallelization (also partitioned in the vertical direction) is applied. The HIP version of LICOM3 (LICOM3-HIP) is 42 times faster than the same number of CPU cores when 384 AMD GPUs and CPU cores are used. LICOM3-HIP has excellent scalability; it can still obtain a speedup of more than four on 9216 GPUs compared to 384 GPUs. In this phase, we successfully performed a test of 1/20° LICOM3-HIP using 6550 nodes and 26200 GPUs, and on a large scale, the model's speed was increased to approximately 2.72 simulated years per day (SYPD). By putting almost all the computation processes inside GPUs, the time cost of data transfer between CPUs and GPUs was reduced, resulting in high performance. Simultaneously, a 14-year spin-up integration following phase 2 of the Ocean Model Intercomparison Project (OMIP-2) protocol of surface forcing was performed, and preliminary results were evaluated. We found that the model results had little difference from the CPU version. Further comparison with observations and lower-resolution LICOM3 results suggest that the 1/20° LICOM3-HIP can reproduce the observations and produce many smaller-scale activities, such as submesoscale eddies and frontal scale structures.

## 1 Introduction

Numerical models are a powerful tool for weather forecasts and climate prediction and projection. Creating high-resolution atmospheric, oceanic and climatic models remains a significant scientific and engineering challenge because of the enormous computing, communication, and input/output (IO) involved. Kilometer-scale weather and climate simulation have recently started to emerge (Schär et al., 2020). Due to the considerable increase in computational cost, such models will only work with extreme-scale high-performance computers and new technologies.

Global ocean general circulation models (OGCMs) are a fundamental tool for oceanography research, ocean forecasting, and climate change research (Chassignet et al., 2019). Such model performance is determined mainly by model resolution and subgrid parameterization and surface forcing. The horizontal resolution of global OGCMs has increased to approximately 5-10 km, which is also called eddy-resolving models. Increasing the resolution will significantly improve the simulation of western boundary currents, mesoscale eddies, fronts and jets, and currents in narrow passages (Hewitt et al., 2017). Meanwhile, the ability of an ocean model to simulate the energy cascade (Wang et al., 2019), the air-sea interaction (Hewitt et al., 2017), and the ocean heat uptake (Griffies et al., 2015) will be improved with increasing resolution. All these factors will effectively improve ocean model performance in the simulation and prediction of ocean circulation. Additionally, the latest numerical and observational results show that much smaller eddies (submesoscale eddies with a spatial scale of approximately 5-10 km) are crucial to vertical heat transport in the upper-ocean mixed layer and significant to biological processes (Su et al., 2018). Resolving the smaller-scale processes raises a new challenge for the horizontal resolution of OGCMs, which also demands much more computing resources.

Heterogeneous computing has become a development trend of high-performance computers. In the latest TOP500 supercomputer list released in November 2020, central processing unit (CPU) and graphic processing unit (GPU) heterogeneous machines account for six of the top 10. After the NVIDIA Corporation provided supercomputing techniques on GPUs, an increasing number of ocean models applied these high-performance acceleration methods to conduct weather or climate simulations. Xu et al. (2015) developed POM.gpu, a full GPU solution based on mpiPOM on a cluster, and achieved a 6.8 times energy reduction. Yashiro et al. (2016) deployed the NICAM model on the TSUBAME supercomputer, and the model sustained a double-precision performance of 60 T Flops on 2560 GPUs. Yuan et al. (2020) developed a GPU version of a wave model with 2 V100 cards and obtained a speedup of 10-12 times when compared to the 36 cores of the CPU. Yang et al. (2016) implemented a fully implicit β-plane dynamic model with a 488 m grid spacing on the TaihuLight system and achieved 7.95P Flops. Fuhrer et al. (2018) reported a 2-km regional atmospheric general circulation model (AGCM) test using 4888 GPU cards and obtained simulation performance for 0.043 simulated years per wall clock day (SYPD). Zhang et al. (2020) successfully ported a high-resolution (25 km atmosphere and 10 km ocean) Community Earth System Model in the TaihuLight supercomputer and obtained 1-3.4 SYPD.

Additionally, the AMD company also provides GPU solutions. In general, AMD GPUs use heterogeneous compute compiler (HCC) tools to compile codes, and they cannot use the compute unified device architecture (CUDA) development environments, which provide support for the NVIDIA GPU only. Therefore, due to the wide use and numerous CUDA learning resources, AMD developers must study two kinds of GPU programming skills. AMD's heterogeneous-compute interface for portability (HIP) is an open-source solution to address this problem. It provides a higher-level framework to contain these two types of lower-level development environments, i.e., CUDA and HCC, simultaneously. The HIP code's grammar is similar to that of the CUDA code, and with a simple conversion tool, the code can be compiled and run at CUDA and AMD architects. HCC/OpenACC is more convenient for AMD GPU developers than the HIP, which is popular from the coding viewpoint.

Another reason is that CUDA GPUs currently have more market share. It is believed that an increasing number of codes will be ported to the HIP in the future. However, almost no ocean models use the HIP framework to date.

    This study aims to develop a high-performance OGCM based on LICOM3, which can be run on an AMD GPU architecture using the HIP framework. Here, we will focus on the model's best or fastest computing performance and its practical usage for research and operation purposes. Section 2 is the introduction of the LICOM3 model. Section 3 contains the main optimization

of LICOM3 under HIP. Section 4 covers the performance analysis and model verification. Section 5 is a discussion, and the conclusion is presented in Section 6.

## 2 The LICOM3 model and experiments

### 2.1 The LICOM3 model

    In this study, the targeting model is the LASG/IAP Climate System Ocean Model version 3 (LICOM3), which was developed

in the late 1980s (Zhang and Liang, 1989). Currently, LICOM3 is the ocean model for two air-sea coupled models of CMIP6, the Flexible Global Ocean-Atmosphere-Land System model version 3 with a finite-volume atmospheric model (FGOALS-f3; He et al., 2020) and the Flexible Global Ocean-Atmosphere-Land System model version 3 with a grid-point atmospheric model (CAS FGOALS-g3; Li et al., 2020). LICOM version 2 (LICOM2.0, Liu et al., 2012) is also the ocean model of the CAS Earth System Model (CAS-ESM, Zhang, et al., 2020). A future paper to fully describe the new features and baseline performances

of LICOM3 is in preparation.

    In recent years, the LICOM model was substantially improved based on LICOM2.0 (Liu et al., 2012). There are three main aspects. First, the coupling interface of LICOM has been upgraded. Now, NCAR flux coupler version 7 is employed (Lin et al., 2016), in which memory usage has been dramatically reduced (Craig et al., 2012). This makes the coupler suitable for application to high-resolution modeling.

Second, both orthogonal curvilinear coordinates (Murray, 1996; Madec & Imbard, 1996) and tripolar grids have been introduced in the LICOM. Now, the two poles are at (65°E, 60.8°N) and (115°W, 60.8°N) for the 1° model, at (65°E, 65°N) and (115°W, 65°N) for the 0.1° model, and at (65°E,60.4°N) and (115°W, 60.4°N) for the 1/20° model of the LICOM. After that, the zonal filter in high latitudes, particularly in the Northern Hemisphere, was eliminated, which significantly improved the scalability and efficiency of the parallel algorithm of the LICOM3 model. In addition, the dynamic core of the model has

also been updated accordingly (Yu et al., 2018), including application of a new advection scheme for the tracer formulation (Xiao, 2006) and addition of a vertical viscosity for the momentum formulation (Yu et al., 2018).

    Third, the physical package has been updated, including introducing an isopycnal and thickness diffusivity scheme (Ferreira et al., 2005) and vertical mixing due to internal tides breaking at the bottom (St. Laurent et al., 2002). The coefficient of both isopycnal and thickness diffusivity is set to $300 m^2 s^{-1}$ as the depth is either within the mixed layer or the water depth is

95 shallower than 60 m. The upper and lower boundary values of the coefficient are 2000 and $300 \ m^2 s^{-1}$, respectively.

Additionally, the chlorophyll-dependent solar shortwave radiation penetration scheme of Ohlmann (2003), the isopycnal mixing scheme (Redi, 1982; Gent & McWilliams, 1990), and the vertical viscosity and diffusivity schemes (Canuto et al. 2001; 2002) are employed in LICOM3.

Both the low-resolution (1°; Lin et al., 2020) and high-resolution (1/10°; Li Y. et al., 2020) stand-alone LICOM3 are also involved in OMIP-1 and OMIP-2; their outputs can be downloaded from websites. The two versions of LICOM3's performances compared with other CMIP6 ocean models are shown in Tsujino et al. (2020) and Chassignet et al. (2020). The 1/10° version has also been applied to perform short-term ocean forecasts (Liu et al., 2021).

The essential task of the ocean model is to solve the approximated Navier-Stocks equations, along with the conservation equations of the temperature and salinity. Seven kernels are within the time integral loop, named "readyt", "readyc", "barotr", "bclinc", "tracer", "icesnow", and "convadj", which are also the main subroutines porting from the CPU to the GPU. The former two kernels computed the terms in the barotropic and baroclinic equations of the model. The following three ("barotr", "bclinc", and "tracer") are used to solve the barotropic, baroclinic, and temperature/salinity equations. The last two subroutines deal with seaice and deep convection processes at high latitudes. All these subroutines have approximately 12000 lines of source code, accounting for approximately 25% of the total code and 95% of computation.

## 2.2 Configurations of the models

To investigate the GPU version, we employed three configurations in the present study. They are 1°, 0.1°, and 1/20°. Details of these models are listed in Table 1. The number of horizontal grid points for the three configurations are $360 \times 218$, $3600 \times 2302$, and $7200 \times 3920$. The vertical levels for the low-resolution models are 30, while they are 55 for the other two high-resolution models. From 1° to 1/20°, the computational effort increased by approximately 8000 ($20^3$) times (considering 20 times to decrease the time step), and the vertical resolution increased from 30 to 55, in total, approximately 15000 times. The original CPU version of 1/20° with MPI parallel on Tianhe-1A only reached 0.31 SYPD using 9216 CPU cores. This speed will slow down the 10-year spin-up simulation of LICOM3 to more than one month, which is not practical for climate research. Therefore, such simulations require extreme-scale high-performance computers by applying the GPU version.

In addition to the different grid points, three main aspects are different among the three experiments, particularly between version 1° and the other two versions. First, the horizontal viscosity schemes are different: using Laplacian for 1° and biharmonic for 1/10° and 1/20°. The viscosity coefficient is one order of magnitude smaller for the 1/20° version than for the 1/10° version, namely, $-1.0 \times 10^9$ m$^4$/s for 1/10° vs $-1.0 \times 10^8$ m$^4$/s for 1/20°. Second, although the force including dataset (JRA55-do; Tsujino et al., 2018) and the bulk formula for the three experiments are all standard of the OMIP-2, the periods and temporal resolutions of the forcing fields are different: 6-hour data from 1958 to 2018 for the 1° version, and daily mean data in 2016 for both the 1/10° and 1/20° versions. Third, version 1° is coupled with a sea ice model of CICE4 via NCAR flux coupler version 7, while the two higher-resolution models are stand-alone, without a coupler or sea ice model. Additionally,

the two higher-resolution experiments employ the new HIP version of LICOM3 (i.e., LICOM3-HIP); the low-resolution experiment does not employ this, including the CPU version of LICOM3 and the version submitted to OMIP (Lin et al., 2020).

We also listed all the important information in Table 1, such as bathymetry data and the bulk formula, though these items are similar in the three configurations.

The spin-up experiments for the two high-resolution versions are conducted for 14 years, forced by the daily JRA55-do dataset in 2016. The atmospheric variables include the wind vectors at 10 m, air temperature at 10 m, relative humidity at 10 m, total precipitation, downward shortwave radiation flux, downward longwave radiation flux, and river runoff. According to the

kinetic energy evolution, the models reach a quasi-equilibrium state after more than ten years of spin-up. The daily mean data are output for storage and analysis.

## 2.3 Hardware and software environments of the testing system

The two higher-resolution experiments were performed on a heterogeneous Linux cluster supercomputer located at the Computer Network Information Center (CNIC) of the CAS, China. This supercomputer consists of 7200 nodes (6 partitions

or rings, each partition has 1200 nodes), with a 1.9 GHz X64 CPU of 32 cores on each node. Additionally, each node is equipped with four gfx906 AMD GPU cards with 16 GB memory. The GPU has 64 cores, for a total of 2560 threads on each card. The nodes are interconnected through high-performance InfiniBand (IB) networks (3-level fat-tree architecture using Mellanox 200 Gb/s HDR InfiniBand, whose measured point-to-point communication performance is approximately 23 GB/s). OpenMPI version 4.02 was employed for compiling, and the AMD GPU driver and libraries were rocm-2.9, integrated with

HIP version 2.8. The storage file system of the supercomputer is ParaStor300S with a 'parastor' file system, whose measured write and read performance is approximately 520 GB/s and 540 GB/s, respectively.

## 3 LICOM3 GPU code structure and optimization

### 3.1 Introduction to HIP on an AMD hardware platform

AMD's HIP is a C++ runtime API and kernel language. It allows developers to create portable applications that can be run on

AMD accelerators and CUDA devices. The HIP provides an API for an application to leverage GPU acceleration for both AMD and CUDA devices. It is syntactically similar to CUDA, and most CUDA API calls can be converted  by replacing the character "cuda" with "hip" (or "Cuda" with "Hip"). The HIP supports a strong subset of CUDA runtime functionality, and its open-source software is currently available on GitHub (https://rocmdocs.amd.com/en/latest/Programming_Guides/HIP-GUIDE.html).

Some supercomputers install NVIDIA GPU cards, such as P100 and V100, and some install AMD GPU cards, such as AMD VERG20. Hence, our HIP version LICOM3 can adapt and gain very high performance at different supercomputer centers, such as Tianhe-2 and AMD clusters. Our coding experience on an AMD GPU indicates that the HIP is a good choice for high-

performance model development. Meanwhile, the model version is easy to keep consistent in these two commonly used platforms. In the following sections, the successful simulation of LICOM3-HIP is confirmed adequate to employ HIP.

Figure 1 demonstrates the HIP implementations necessary to support different types of GPUs. In addition to the differences in naming and libraries, there are other differences between HIP and CUDA including the following: 1) AMD Graphics Core Next (GCN) hardware "warp" size = 64; 2) device and host pointers allocated by HIP API use flat addressing (unified virtual addressing is enabled by default); 3) dynamic parallelism is not currently supported; 4) some CUDA library functions do not have AMD equivalents; and 5) shared memory and registers per thread may differ between the AMD and NVIDIA hardware.

Despite these differences, most of the CUDA codes in applications can be easily translated to the HIP and vice versa. Technical supports of CUDA and HIP also have some differences. For example, CUDA applications have some CUDA-aware MPI to direct MPI communication between different GPU memory spaces at different nodes, but HIP applications have no such functions to date. Data must be transferred from GPU memory to CPU memory in order to exchange data with other nodes and then transfer data back to the GPU memory.

**3.2. Core computation process of LICOM3 and C transitional version**

We attempted to apply the LICOM on a heterogeneous computer approximately five years ago, cooperating with the NVIDIA Corporation. LICOM2 was adapted to NVIDIA P80 by OpenACC Technical (Jiang et al., 2019). That was a convenient implementation of LICOM2-gpu using 4 NVIDIA GPUs to achieve a 6.6 speedup compared to 4 Intel CPUs, but its speedup was not as good when further increasing the GPU number.

During this research, we started from the CPU version of LICOM3. The code structure of LICOM3 includes four steps. The first step is the model setup, which involves MPI partitioning and ocean data distribution. The second stage is model initialization, which includes reading the input data and initializing the variables. The third stage is integration loops, or the core computation of the model. Three explicit time loops, which are for tracer, baroclinic and barotropic steps, are in one model day. The outputs and final processes are included in the fourth step.

Figure 2 shows the flowchart of LICOM3. The major processes within the model time integration include baroclinic, barotropic, and thermohaline equations, which are solved by the leapfrog or Euler forward scheme. There are seven individual subroutines, such as "readyt", "readyc", "barotr", "bclinc", "tracer", "icesnow", and "convadj". When the model finishes one day's computation, the diagnostics and output subroutine will write out the predicted variables to files. The output files contain all the necessary variables to restart the model and for analysis.

To obtain high performance, it is more efficient to use the native GPU development language. In the CUDA development forum, both CUDA-C and CUDA-Fortran are provided; however, Fortran's support is not as efficient as that for C++. We plan to push all the core process codes into GPUs; hence, the seven significant subroutines' Fortran codes must be converted to HIP/C++. Due to the complexity and many lines in these subroutines (approximately 12000 lines of Fortran code) and to ensure that the converted C/C++ codes are correct, we rewrote them to C before finally converting them to HIP codes.

A bit-reproducible climate model produces the same numerical results for a given precision, regardless of the choice of domain decomposition, the type of simulation (continuous or restart), compilers, and the architectures executing the model (i.e., the same hardware and software conduct the same result). The C transitional version (not fully C code, but the seven core subroutines) is bit reproducible with the F90 version of LICOM3 (the binary output data are the same under Linux with the "diff" command). We also tested the execution time. The Fortran and C hybrid version's speed is slightly faster (less than 10%)

than the original Fortran code. Figure 3 shows a speed benchmark by LICOM3 for 100 km and 10 km running on an Intel platform. The results are the wall clock time of running one model month for a low-resolution test and one model day for a high-resolution test. The details of the platform are in the caption of Figure 3. The results indicate that we successfully ported these kernels from Fortran to C.

This C transitional version becomes the starting point of HIP/C++ codes and reduces the complexity of developing the HIP

version of LICOM3.

### 3.3. Optimization and tuning methods in LICOM3-HIP

The unit of computation in LICOM3-HIP is a horizontal grid point. For example, $1/20°$ corresponds to $7200\times3920$ grids. For the convenience of MPI parallelism, the grid points are grouped as blocks; that is, if $Proc_x\times Proc_y$ MPI processes are used in the x and y directions, then each block has $B_x\times B_y$ grids, where $Proc_x\times B_x=7200$ and $Proc_y\times B_y=3920$. Each GPU process performs

2-dimensional (2-D) or 3-dimensional (3-D) computations in these $B_x\times B_y$ grids, which is similar to the MPI process. 2-D means that the grids are partitioned only in the horizontal directions, and 3-D includes also the depth or vertical direction. In practice, four lateral columns are added to $B_x$ and $B_y$ (two on each side, $imt=B_x+4$, $jmt=B_y+4$) for the halo. Table 2 lists the frequently used block definitions of LICOM3.

The original LICOM3 was written in F90. To adapt it to a GPU, we applied Fortran/C hybrid programming. As shown in

Figure 2, the codes are kept using the F90 language before entering device-stepon and after stepon-out. The core computation processes within the stepons are rewritten using HIP/C. Data structures in the CPU space remain the same as the original Fortran structures. The data commonly used by F90 and C are then defined by extra C, including files and defined by "extern" type pointers in C syntax to refer to them. In the GPU space, newly allocated GPU global memories hold the arrival correspondence to those in the CPU space, and the HipMemcpy is called to copy them in and out.

Seven major subroutines (including their subrecurrent calls) are converted from Fortran to HIP. The seven subroutine call sequences are maintained, but each subroutine is deeply recoded in the HIP to obtain the best performance. The CPU space data are 2-D or 3-D arrays; in the GPU space, they are changed to 1-D arrays to improve the data transfer speed between different GPU subroutines.

The LICOM3-HIP is two-level parallelism, and each MPI process corresponds to an ocean block. The computation within one

MPI process is then pushed into the GPU. The latency of the data copy between the GPU and CPU is one of the bottlenecks

for daily computation loops. All read-only GPU variables are allocated and copied at the initial stage to reduce the data copy time. Some datum copy is still needed in the stepping loop, e.g., MPI call in barotr.cpp.

The computation block in MPI (corresponding to 1 GPU) is a 3-D grid; in the HIP revision, 3-D parallelism is implemented. This change adds more parallel inside one block than the MPI solo parallelism (only 2-D). Some optimizations are needed to adapt to this change, such as increasing the global arrays to avoid data dependency. A demo for using a temporary array to parallelize the computation inside a block is shown in Figure 4. Figure 4a represents a loop of the original code in the $k$ direction. Since the variable $v(i,j,k)$ has a dependence on $v(i,j,k+1)$, it will cause an error when the GPU threads are parallel in the $k$ direction. We then separate the variable into two HIP kernel computations. In the upper part of Figure 4b, a temporary array $vt$ is used to hold the result of $f1()$, and it can be GPU threads that are parallel in the $k$ direction. Then, at the bottom of Figure 4b, we use $vt$ to perform the computations of $f2()$ and $f3()$; this can still be GPU threads that are parallel in the $k$ direction. Finally, this loop of codes is parallelized.

Parallelization in a GPU is similar to a shared-memory program; memory write conflicts occur in the subroutine "tracer" advection computation. We change the if-else tree in this subroutine; hence, the data conflicts between neighboring grids are avoided, making the 3-D parallelism successful. Moreover, in this subroutine, we use more operations to alternate the data movement to reduce the cache usage. Since the operation can be GPU thread parallelized and will not increase the total computation time, reducing the memory cache improves this subroutine's final performance.

A notable problem when the resolution is increased to $1/20°$ is that the total size of Fortran common blocks will be larger beyond 2 GB. This change will not cause abnormalities for C in the GPU space. However, if the GPU process references the data, the system call in HipMemcpy will cause compilation errors (perhaps due to the compiler limitation of the GPU compilation tool). We can change the original Fortran arrays' data structure from the "static" to the "allocatable" type in this situation. Since a GPU is limited to 16 GB GPU memory, the ocean block size in one block should not be too large. In practice, the $1/20°$ version starts from 384 GPUs (and is regarded as the baseline for speedup here); if the partition is smaller than that value, sometimes insufficient GPU memory errors will occur.

We found that the "tracer" is the most time-consuming subroutine for the CPU version (Figure 5). With the increase of CPU cores from 384 to 9216, the ratio of cost time for "tracer" is also increasing from 38% to 49%. "readyt" and "readyc" are computing-intensive subroutines. "Tracer" is both a computing-intensive and communication-intensive subroutine. "barotr" is a communication-intensive subroutine. The communication of "barotr" is 45 times more than that of "tracer" (Table 3). Computing-intensive subroutines can achieve good GPU speed, but communication-intensive subroutines will achieve poor performance. The superlinear speedups for "tracer" and "readyc" might be mainly caused by memory usage, in which the memory usage of each thread for 768 GPU cards is only half for 384 GPU cards.

We performed a set of experiments to measure the time cost of both halo update and memory copy in the HIP version (Figure 6). These two processes in the time integration are conducted in three subroutines: "barotr", "bclinc," and "tracer". The figure shows that "barotr" is the most time-consuming subroutine, and the memory copy dominates, which takes approximately 40% of the total time cost.

Data operations inside CPU (or GPU) memory are at least one order of magnitude faster than the data transfer between GPU and CPU through 16X PCI-e. Halo exchange at the MPI level is similar to POP (Jiang et al. 2019). We did not change these codes in the HIP version. The four blue rows and columns in Figure 7 demonstrate the data that need to be exchanged with the neighbors. As shown in Figure 7, in GPU space, we pack the necessary lateral data for halo operation from $imt \times jmt$ to $4(imt+jmt)$. This change reduces the HipMemcpy data size to $(4/imt+4/jmt)$ of the original one. The larger that $imt$ and $jmt$ are, the less the transferred data are. At 384 GPUs, this change saves approximately 10% of the total computation time. The change is valuable for the HIP since the platform has no CUDA-aware MPI installed; otherwise, the halo operation can be done in the GPU space directly as done by POM.gpu (Xu et al., 2015). The test indicates that the method can decrease approximately 30% of the total wall clock time of "barotr" when 384 GPUs are used. However, we have not optimized other kernels so far because their performance is not as good as 384 GPUs when the GPU scale exceeds 10000. We keep it here as an option to improve the performance of 'barotr' at operational scales (i.e., GPU scales under 1536).

### 3.4. Model I/O optimization

Approximately 3 GB forcing data are read from the disk every model year, while approximately 60 GB daily mean predicted variables are stored to disk every model day. The time cost for reading daily forcing data from the disk increased to 200 s in one model day after the model resolution increased from 1° to 1/20°. This time is comparable to the wall clock time for one model step when 1536 GPUs are applied; hence, we must optimize the model for total speedup. The cause of low performance is daily data reading and scattering to all nodes every model day; we then rewrite the data reading strategy and perform parallel scattering for ten different forcing variables. Originally, 10 variables are read from 10 files, interpolated to 1/20° grid and then scattered to each processor or thread. All the processes are sequentially done at the master processor. In the revised code, we use 10 different processes to read, interpolate and scatter parallelly. Finally, the time cost of input is reduced to approximately 20 s, which is 1/10 of the original time cost (shown below).

As indicated, the time cost for one integration step (excluding the daily mean and I/O) is approximately 200 s using 1536 GPUs. One model day's output needs approximately 250 s; this is also beyond the GPU computation time for one step. We modify the subroutine to a parallel version, which decreases the data write time to 70 s on the test platform (this also depends on system I/O performance).

### 4 Model performance

#### 4.1. Model performance in computing

Performing kilometer-scale and global climatic simulations is challenging (Palmer, 2014; Schär et al., 2020). As specified by Fuhrer et al. (2018), the SYPD is a useful metric to evaluate model performance for a parallel model (Balaji et al., 2017). Because a climate model often needs to run for at least 30-50 years for each simulation, at a speed of 0.2-0.3 SYPD, the time

will be too long to finish the experiment. The common view is that at least 1-2 SYPD is an adequate entrance for a realistic climate study. It also depends on the time scale in a climate study. For example, for the 10-20-year simulation, 1-2 SYPD seems acceptable, and for the 50-100-year simulation, 5-10 SYPD is better. The NCEP weather prediction system throughput standard is 8 minutes to finish one model day, equivalent to 0.5 SYPD.

Figure 8 illustrates the I/O performance of LICOM3-HIP, comparing the performances of computation processes. When the
model applies 384 GPUs, the I/O costs 1/10 of the total simulation time (Figure 8a). When the scale increases to 9216 GPUs, the I/O time increases but is still smaller than the GPU's step time (Figure 8b). The improved LICOM3 I/O in total costs approximately 50-90 s (depending on scales), especially when the input remains stable (Figure 8c) while scaling increases. This optimization of I/O maintains that LICOM3-HIP 1/20° runs well at all practice scales for a realistic climate study. The I/O time was cut off from the total simulation time in the follow-up test results to analyze the purely parallel performance.

Figure 9 shows the roof-line model using the Stream-GPU and the LICOM program's measured behavioral data on a single computation node bound to one GPU card depicting the relationship between arithmetic intensity and performance floating point operations. The 100 km resolution case is employed for the test. The blue and gray oblique lines are the fitting lines related to the Stream-GPU program's behavioral data using 5.12e8 and 1e6 threads, respectively, both with a blocksize of 256, which attain the best configuration. For details, the former is approximately the maximum thread number restricted by GPU
card memory, achieving the bandwidth limit to 696.52 GB/s. In comparison, the latter is close to the average number of threads in GPU parallel calculations used by LICOM, reaching a bandwidth of 344.87 GB/s on average. Here, we use the oblique gray line as a benchmark to verify the rationality of LICOM's performance, accomplishing an average bandwidth of 313.95 GB/s. Due to the large calculation scale of the entire LICOM program, the divided calculation grid bound to a single GPU card is limited by video memory; most kernel functions issue no more than 1.2e6 threads. As a result, the floating-
point operation performance is slightly far from the oblique roof-line shown in Figure 9. In particular, the subroutine bclinc apparently strays off of the entire trend for including frequent 3d-array Halo MPI communications and much data transmission occurs between the CPU and GPU.

Figure 10 shows the SYPD at various parallel scales. The baseline (384) of GPUs could achieve a 42 time speedup than that of the same number of CPU cores. Sometimes, we also count the overall speedup, 384 GPUs in 96 nodes versus the total 3072
CPU cores in 96 nodes. We can obtain an overall performance speedup of 384, or approximately 6-7 times. The figure also indicates that for all scales, the SYPD continues to increase. On the scale of 9216 GPUs, the SYPD first goes beyond 2, which is seven times the same CPU result. A quasi-whole machine (26200 GPUs, 26200×65=1703000 cores in total, one process corresponds to one CPU core plus 64 GPU cores) result indicates that it can still obtain an increasing SYPD to 2.72.

Since each node has 32 CPU cores and 4 GPUs, each GPU is managed by one CPU thread in the present cases. We can also
quantify GPUs' speedup vs. all CPU cores' on the same number of nodes. For example, the 384 (768) GPUs correspond to 96 (192) nodes, which have 3072 (6144) CPU cores. Therefore, the overall speedup is approximately 6.375 (0.51/0.08) for 384 GPUs and 4.15 (0.83/0.2) for 768 GPUs (Figure 10). The speedups are comparable with our previous work porting LICOM2 to GPU using OpenACC (Jiang et al., 2019), which is approximately 1.8-4.6 times the speedup using one GPU card vs. two

8-core Intel GPU in small-scale experiments for specific kernels. Our results are also slightly better than Xu et al. (2015), who ported another ocean model to GPUs using Cuda C. However, due to the limitation of the number of intel CPUs (maximal 9216 cores), we did not obtain the overall speedup for 1536 and more GPUs.

Figure 11 depicts the actual times and speedups of different GPU computations. The green line in Figure 11a is a function of the stepon time cost; it decreases while the GPU number increases. The blue curve of Figure 11a shows the increase in speedup with the rise in the GPU scale. Despite the speedup increase, the efficiency of the model decreases. At 9216 GPUs, the model efficiency starts under 20%, and for more GPUs (19600 and 26200), the efficiency is flattened to approximately 10%. The efficiency decrease is mainly caused by the latency of the data copy in and out to the GPU memory. For economic consideration, the 384-1536 scale is a better choice for realistic modeling studies.

Figure 12 depicts the time cost of seven core subroutines of LICOM3-HIP. We find that the top four most time cost subroutines are "barotr," "tracer," "bclinc," and "readyc", and the other subroutines cost only approximately 1% of the whole computation time. When 384 GPUs are applied, the "barotr" costs approximately 50% of the total time (Figure 12a), which solves the barotropic equations. When GPUs are increased to 9216, each subroutine's time cost decreases, but the percentage of subroutine "barotr" is increased to 62% (Figure 12b). As mentioned above, this phenomenon can be interpreted by having more haloing in "barotr" than in the other subroutines; hence, the memory data copy and communication latency make it slower.

## 4.2. Model performance in climate research

The daily mean sea surface height (SSH) fields of the CPU and HIP simulations are compared to test the usefulness of the HIP version of LICOM for the numerical precision of scientific usage. Here, the results from 1/20° experiments on a particular day, March 1st of the 4th model year, are used (Figures 13a, b). The general SSH spatial patterns of the two are visually very similar. Significant differences are only found in very limited areas, such as in the eddy-rich regions near strong currents or high-latitude regions (Figure 13c); in most places, the difference in values fall into the range of -0.1 and 0.1 cm. Because the hardware is different and the HIP codes' mathematical operation sequence is not always the same as that for the Fortran version, the HIP and CPU versions are not identical byte-by-byte. Therefore, it is hard to verify the correctness of the results from the HIP version. Usually, the ensemble method is employed to evaluate the consistency of two model runs (Baker et al., 2015). Considering the unacceptable computing and storage resources, in addition to the differences between the two versions, we simply compute root mean square errors (RMSEs) between the two versions, which are only 0.18 cm, much smaller than the spatial variation of the system, which is 92 cm (approximately 0.2%). This indicates that the results of LICO3-HIP are generally acceptable for research.

The GPU version's sea surface temperature (SST) is compared with the observed SST to evaluate the global 1/20° simulation's preliminary results from LICOM3-HIP (Figure 14). Because the LICOM3-HIP experiments are forced by the daily mean atmospheric variables in 2016, we also compare the outputs with the observation data in 2016. Here, the 1/4° Optimum Interpolation Sea Surface Temperature (OISST) is employed for comparison, and the simulated SST is interpolated to the same resolution as the OISST. We find that the global mean values of SST are close together, but with a slight warming bias of

18.49°C for observations vs. 18.96°C for the model. The spatial pattern of SST in 2016 is well reproduced by LICOM3-HIP. The spatial standard deviation (STD) of SST is 11.55°C for OISST and 10.98°C for LICOM3-HIP. The RMSE of LICOM3-HIP against the observation is only 0.84°C.

355 With an increasing horizontal resolution of the observations, we now know that mesoscale eddies are ubiquitous in the ocean at the 100-300 km spatial scale. Rigorous eddies usually occur along significant ocean currents, such as the Kuroshio and its extension, the Gulf Stream, and the Antarctic Circumpolar Current (Figure 15a). Eddies also capture more than 80% of the ocean's kinetic energy, which was estimated using satellite data (e.g., Chelton et al., 2011). Therefore, these mesoscale eddies must be solved in the ocean model. A numerical model's horizontal resolution must be higher than 1/10° to resolve the global

ocean eddies but cannot resolve the eddies in high latitude and shallow waters (Hallberg, 2013). Therefore, a higher resolution is required to determine the eddies globally. The EKE for the 1° version is low, even in the areas with strong currents, while the 1/10° version can reproduce most of the eddy-rich regions in the observation. The EKE increases when the resolution is further enhanced to 1/20°, indicating that many more eddy activities are resolved.

## 5 Discussion

**5.1. Application of the ocean climate model beyond 10000 GPUs**

Table 4 summarizes the detailed features of some published GPU version models. We find that various programming methods have been implemented for different models. A near-kilometer atmospheric model using 4888 GPUs was reported as a large-scale example of weather/climate studies. With supercomputing development, the horizontal resolution of ocean circulation models will keep increasing, and more sophisticated physical processes will also be developed. The LICOM3-HIP has a larger

scale, not only in terms of grid size but also in final GPU numbers.

We successfully performed a quasi-whole machine (26200 GPUs) test, and the results indicate that the model obtained an increasing SYPD (2.72). The application of an ocean climate model beyond 10000 GPUs is not easy because the multinodes plus multi-GPUs running requires that the network connection, PCI-e and memory speed, and input/output storage systems all work in their best performances. Gupta et al. (2017) investigated 23 types of system failures to improve reliability of the HPC

system. Unlike Gupta's study, only the three most common types of failures we encountered are discussed here. The three most common errors when running LICOM3-HIP are MPI hardware errors, CPU memory access errors, and GPU hardware errors. Let us suppose that the probability of an individual hardware (or software) error occurring is $10^{-5}$ (which means 1 failure in 100000 hours). As the MPI (GPU) scale increases, the total error rate increases, and once a hardware error occurs, the model simulation will fail.

When 384 GPUs are applied, the success rate within one hour can be expressed as $\left(1\text{-}384\times10^{-5}\right)^3$=98.85%, and the failure rate is then $1\text{-}\left(1\text{-}384\times10^{-5}\right)^3$=1.15%. Applying this formula, we can obtain the failure rate corresponding to 1000, 10000, and 26200 GPUs. The results are listed in Table 5. As shown in Table 5, on the medium scale (i.e., 1000 GPUs are used), three

failures will occur through 100 runs; when the scale increases to 10000 GPUs, 1/4 of them will fail. The $10^{-5}$ error probability also indicates that 10000 GPU tasks cannot run ten continuous hours on average. If the success time restriction decreases, the model success rate will increase. For example, within 6 minutes, the 26200 GPU task success rate is $(1\text{-}26200\times10^{-6})^3$=92.34%, and its failure rate is $1\text{-}(1\text{-}26200\times10^{-6})^3$=7.66%.

## 5.2. Energy to solution

We also measured energy to solution here. A simulation normalized energy (E) is employed here as a metric. The formula is as follows:

$$E = TDP \times N \times 24/SYPD$$

where TDP is the thermal design power, N is the computer nodes used, and SYPD/24 equals the simulated years per hour. Therefore, the smaller the E value is, the better, which means that we can obtain more simulated years within a limited power supply. To calculate E's value, we estimated the TDP of 1380 W for a node on the present platform (1 AMD CPU and 4 GPUs) and 290 W for a reference node (2 Intel 16-core CPUs). We only include the TDP of CPUs and GPUs here.

Based on the above power measurements, simulations' energy cost is shown in Table 6 in MWh per simulation year (MWh/SY). The energy costs for the 1/20° LICOM3 simulations running on CPUs and GPUs are comparable when the numbers of MPI processors are within 1000. The energy costs of LICOM3 at 1/20° running on 384 (768) GPUs and CPUs are approximately 6.234 (7.661) MWh/SY and 6.845 (6.280) MWh/SY, respectively. However, the simulation speed of LICOM3 on a GPU is much faster than that on a CPU, approximately 42 times for 384 processors and 31 times for 768 processors. When the number of MPI processors is beyond 1000, the value of E for the GPU becomes much larger than that for the CPU. This result indicates that the GPU is not fully loaded at this scale.

## 6 Conclusion

The GPU version of LICOM3 under the HIP framework was developed in the present study. Seven kernels within the time integration of the mode are all ported to the GPU, and 3-D parallelization (also partitioned in the vertical direction) is applied. The new model was implemented and gained an excellent acceleration rate on a Linux cluster with AMD GPU cards. This is also the first time an ocean general circulation model has been fully applied on a heterogeneous supercomputer using the HIP framework. It totally took nineteen months, five Ph.D. students and five part-time staff to finish the porting and testing work. Based on our test using the 1/20° configuration, LICOM3-HIP is 42 times faster than the CPU when 384 AMD GPUs and CPU cores are used. LICOM3-HIP has good scalability, and can obtain a speedup of more than four on 9216 GPUs compared to 384 GPUs. The SYPD, which is in equilibrium with the speedup, continues to increase as the number of GPUs increases. We successfully performed a quasi-whole machine test, which was 6550 nodes and 26200 GPUs, using 1/20° LICOM3-HIP on the supercomputer, and at the grand scale, the model can obtain an increasing SYPD of 2.72. The modification or

optimization of the model also improves the 10- and 100-km performances, although we did not analyze their performances in this article.

The efficiency of the model decreases with the increasing number of GPUs. At 9216 GPUs, the model efficiency starts under 20% against 384 GPUs, and when the number of GPUs reaches or exceeds 20000, the efficiency is only approximately 10%. Based on our kernel functions test, the decreasing efficiency was mainly caused by the latency of data copy in and out to the GPU memory in solving the barotropic equations, particularly for the number of GPUs larger than 10000.

Using the 1/20° configuration of LICOM3-HIP, we conducted a 14-year spin-up integration. Because the hardware is different

and the GPU codes' mathematical operation sequence is not always the same as that of the Fortran version, the GPU and CPU versions cannot be identical byte by byte. The comparison between the GPU and CPU versions of LICOM3 shows that the differences in most places are minimal, indicating that the results from LICOM3-HIP can be used for practical research. Further comparison with the observation and the lower-resolution results suggests that the 1/20° configuration of LICOM3-HIP can reproduce the observed large-scale features and produce much smaller-scale activities than that of lower-resolution results.

The eddy-resolving ocean circulation model, which is a virtual platform for oceanography research, ocean forecasting, and climate prediction and projection, can simulate the variations in circulations, temperature, salinity, and sea level with a spatial scale larger than 15 km and temporal scale from the diurnal cycle to decadal variability. As mentioned above, 1-2 SYPD is a good entrance for a realistic climate research model. The more practical GPU scale range for realistic simulation is approximately 384-1536 GPUs. At these scales, the model still has 0.5-1.22 SYPD. Even if we decrease the loops in the

"barotr" procedure to 1/3 of the original in the spin-up simulation, the performance will achieve 1-2.5 SYPD for 384-1536 GPUs. This performance will satisfy 10-50-year scale climate studies. In addition, this version can be used for short-term ocean prediction in the future.

Additionally, the block size of 36×30×55 (1/20° setup, 26200 GPUs) is not an enormous computational task for one GPU. Since one GPU has 64 cores total of 2560 threads, if a subroutine computation is 2-D, each thread's operation is too small.

Even for the 3-D loops, it is still not large enough to load the entire GPU. This indicates that it will gain more speedup when the LICOM resolution is increased to the kilometer level. The LICOM3-HIP codes are now written for 1/20°, but they are kilometer-ready GPU codes.

The optimization strategies here are mostly at the program level and do not treat the dynamic or physics parts separately. We only ported all seven core subroutines within the time integration loops to the GPU, including both the dynamic and physics

parts. Unlike atmospheric models, there are few time-consuming physical processes in ocean models, such as radiative transportation, clouds, precipitation, and convection processes. Therefore, the two kinds of parts are usually not separated in the ocean model, particularly in the early stage of model development. This is also the case for LICOM. Further optimization to explicitly separate the dynamic core and the physical package is necessary in the future.

There is still potential to further increase the speedup of LICOM3-HIP. The bottleneck is in the high-frequency data copy in

and out to the GPU memory in the barotropic part of the LICOM3. Unless HIP-aware MPI is supported, the data transfer latency between the CPU and GPU cannot be overcome. Thus far, we can only reduce the time consumed by decreasing the

frequency or magnitude of the data copy and even modifying the method to solve the barotropic equations. Additionally, using single precision within the time integration of LICOM3 might be another solution. The mixing precision method has already been tested using an atmospheric model, and the average gain in computational efficiency is approximately 40% (Váňa et al., 2017). We would like to try these methods in the future.

## Code availability

The model code (LICOM3-HIP V1.0) along with the dataset and a 100 km case can be downloaded from the website https://zenodo.org/record/4302813#. X8 mGWcsvNb8 with the Digital Object Identifier (doi): 10.5281/zenodo.4302813.

## Data availability

The data for figures in this paper can be downloaded from https://zenodo.org/record/4542544#. YCs24c8vPII with doi: 10.5281/zenodo.4542544.

## Author contribution

Pengfei Wang: Software, visualization, formal analysis, and writing-original draft

Jinrong Jiang: Software and Writing – review & editing

Pengfei Lin: Software and Writing – review & editing

Mengrong Ding: Visualization and Data curation

Junlin Wei: Software

Zhang Feng: Software

Lian Zhao: Software

Yiwen Li: Software and Visualization

Zipeng Yu: Software and Data curation

Weipeng Zheng: Formal analysis

Yongqiang Yu: Conceptualization

Xuebin Chi: Conceptualization

Hailong Liu: Supervision, Formal analysis, and writing-original draft

**Competing interests:**

The authors declare that they have no known competing financial interests or personal relationships that could have appeared to influence the work reported in this paper.

**Acknowledgments**

The study is funded by the National Natural Sciences Foundation (41931183), the National Key Research and Development Program (2018YFA0605904 and 2018YFA0605703), and the Strategic Priority Research Program of the Chinese Academy of Sciences (XDC01040100). Dr.s H.L.L. and P.F.L. were also supported by the "Earth System Science Numerical Simulator Facility" (EarthLab).

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

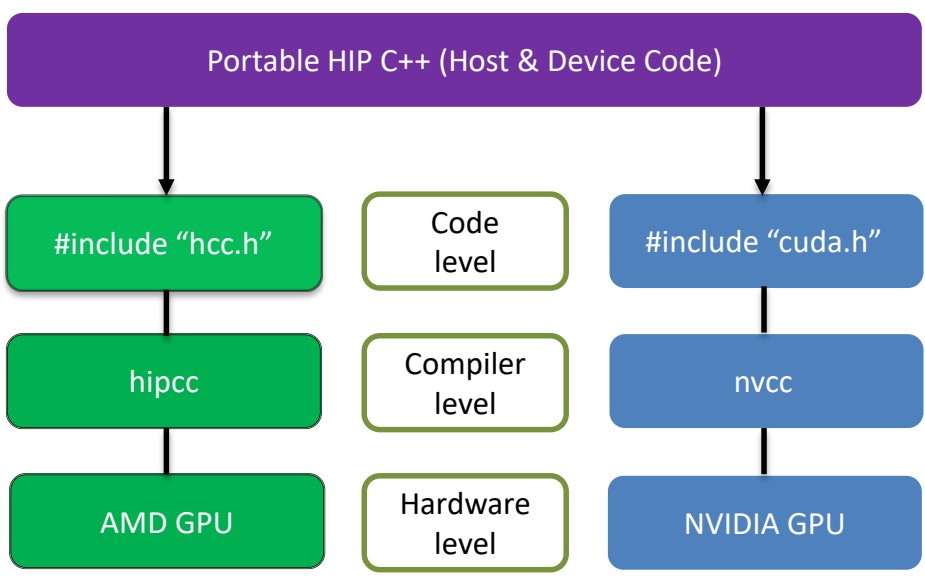

**Figure 1: Schematic diagram of the comparison of coding on AMD and NVIDIA GPUs at three levels.**

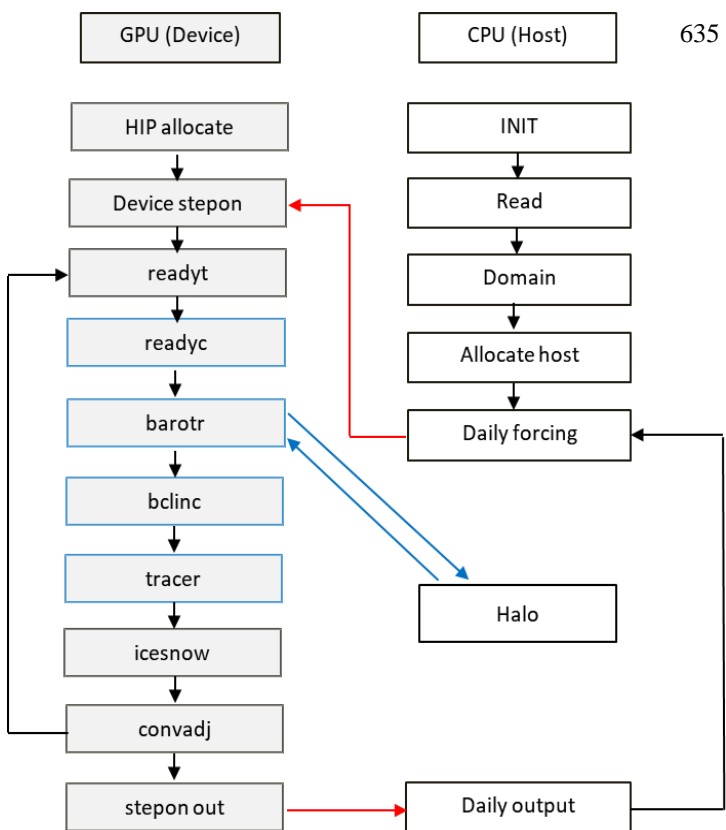

**Figure 2: LICOM3 computation flowchart with a GPU (HIP device). The red line indicates whole block data transfer between the host and GPU, while the blue line indicates transferring only lateral data of a block.**

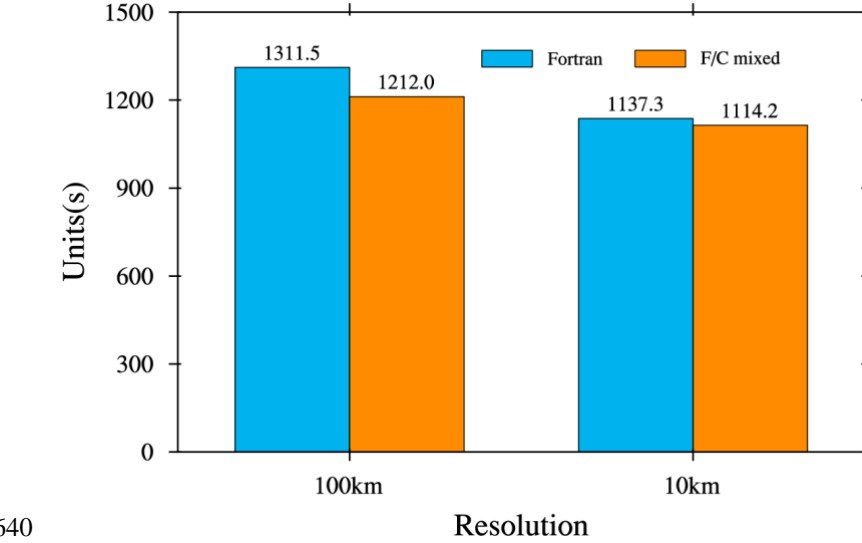


**Figure 3: The wall clock time of a model day for the 10 km version and a model month for the 100 km version. The blue and orange bars are for the Fortran and Fortran and C mixed versions. These tests were conducted on an Intel Xeon CPU platform (E5-2697A v4, 2.60 GHz). We used 28 and 280 cores for the low and high resolutions, respectively.**


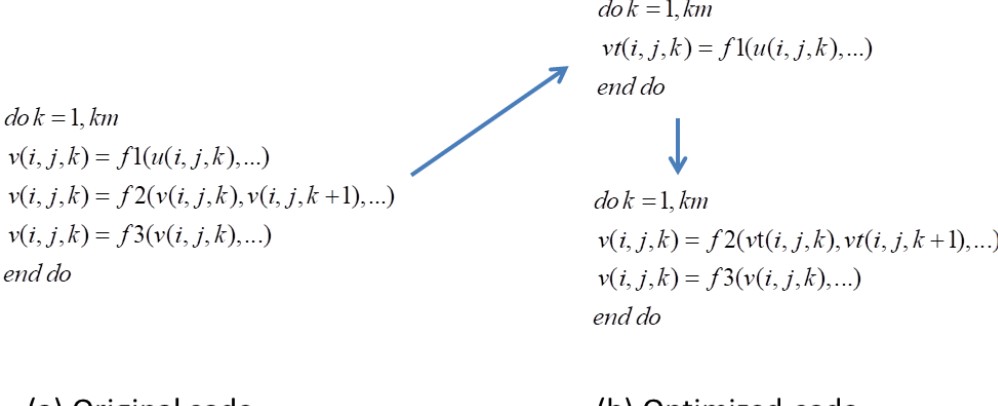

$do\ k = 1, km$

$v(i, j, k) = f1(u(i, j, k), ...)$

$v(i, j, k) = f2(v(i, j, k), v(i, j, k+1), ...)$

$v(i, j, k) = f3(v(i, j, k), ...)$

$end\ do$

$do\ k = 1, km$

$vt(i, j, k) = f1(u(i, j, k), ...)$

$end\ do$

$do\ k = 1, km$

$v(i, j, k) = f2(vt(i, j, k), vt(i, j, k+1), ...)$

$v(i, j, k) = f3(v(i, j, k), ...)$

$end\ do$

(a) Original code                    (b) Optimized code

**Figure 4: The code using temporary arrays to avoid data dependency.**

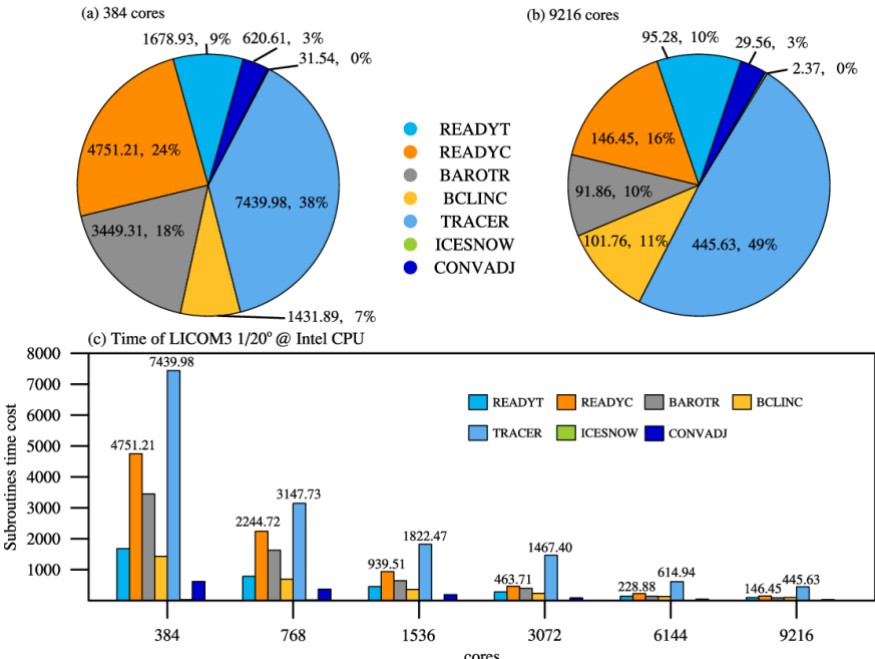


**Figure 5: The seven core subroutines' time cost percentages for (a) 384 and (b) 9216 CPU cores. (c) The subroutines' time cost at different scales of LICOM3 (1/20°). These tests were conducted on an Intel Xeon CPU platform (E5-2697A v4, 2.60 GHz).**

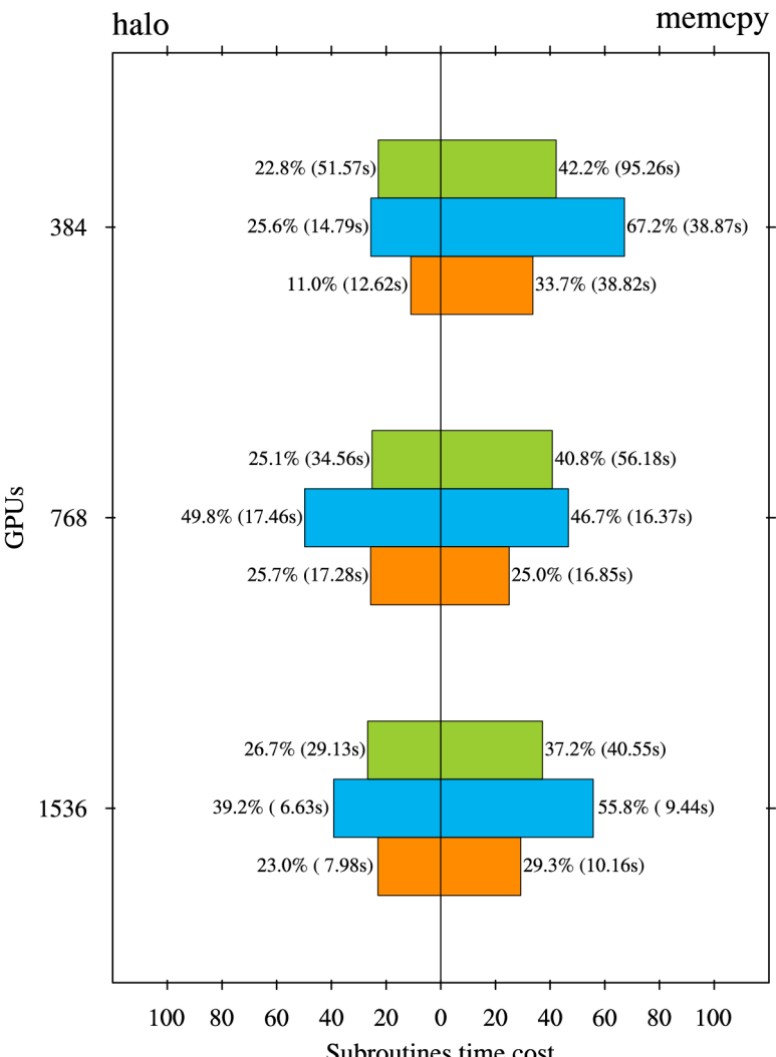

halo        memcpy

**384**
- 22.8% (51.57s) | 42.2% (95.26s)
- 25.6% (14.79s) | 67.2% (38.87s)
- 11.0% (12.62s) | 33.7% (38.82s)

**768**
- 25.1% (34.56s) | 40.8% (56.18s)
- 49.8% (17.46s) | 46.7% (16.37s)
- 25.7% (17.28s) | 25.0% (16.85s)

**1536**
- 26.7% (29.13s) | 37.2% (40.55s)
- 39.2% ( 6.63s) | 55.8% ( 9.44s)
- 23.0% ( 7.98s) | 29.3% (10.16s)

GPUs

Subroutines time cost

**Figure 6: The ratio of the time cost of halo update and memory copy to the total time cost for the three subroutines, "barotr" (green), "bclinc" (blue), and "tracer" (orange), in the HIP version LICOM for three scales (Unit: %). The numbers in the blankets are the time cost of the two processes (unit: second).**

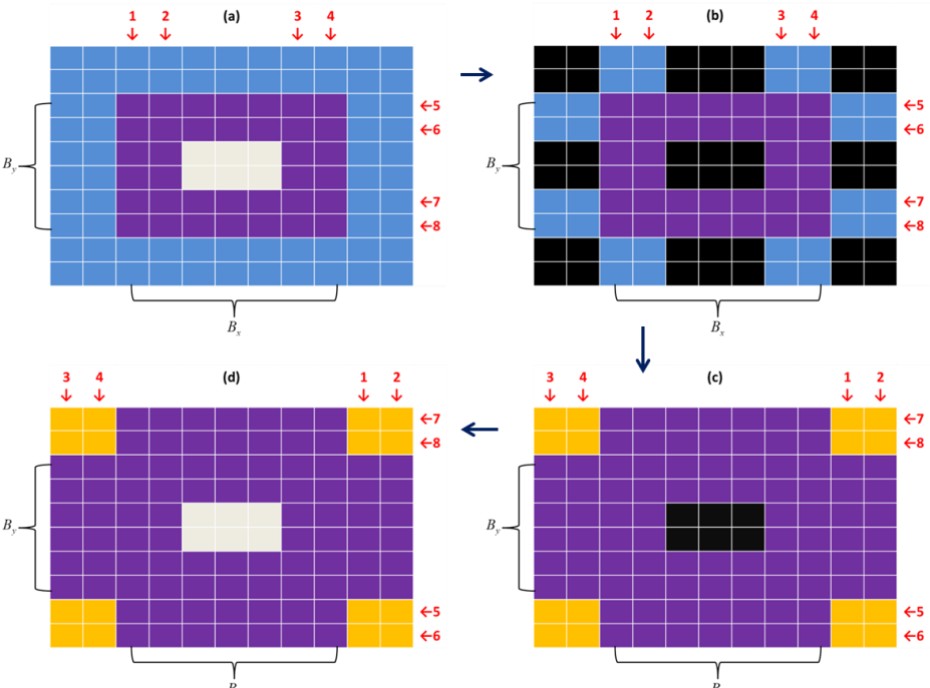

**Figure 7: The lateral packing (only transferring four rows and four columns of data between the GPU and CPU) method to accelerate the halo. (a) In the GPU space, where central (gray) grids are unchanged; (b) transferred to the CPU space, where black grids mean no data; (c) after halo with neighbors; and (d) transfer back to the GPU space.**

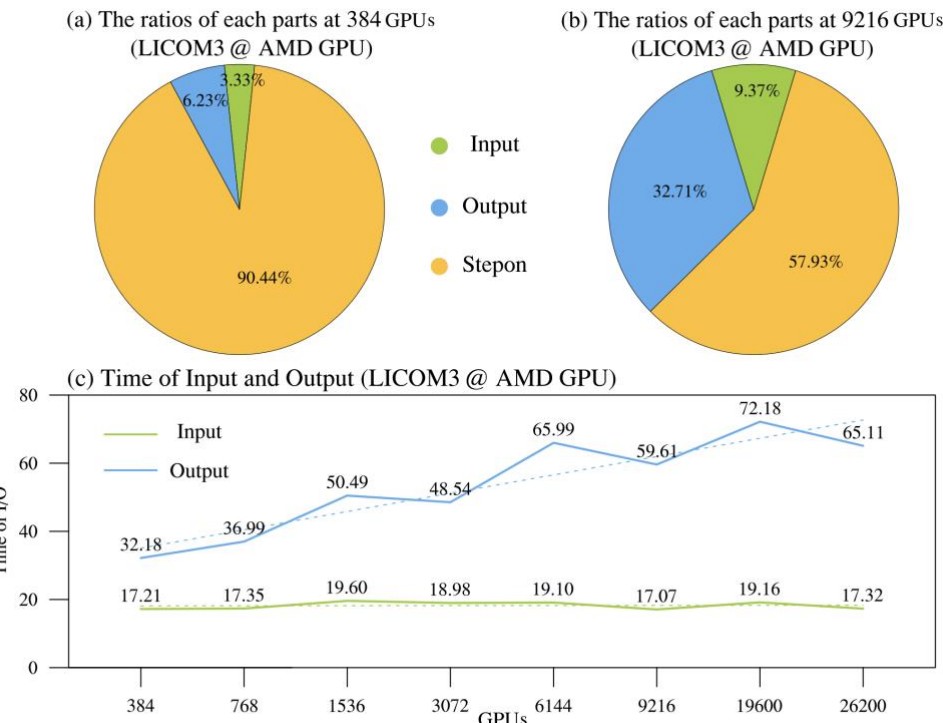

**Figure 8: (a) The 384 GPUs, (b) 9216 GPUs, the I/O ratio in total simulation time for 1/20° setup, and (c) the changes of I/O times versus different GPUs.**

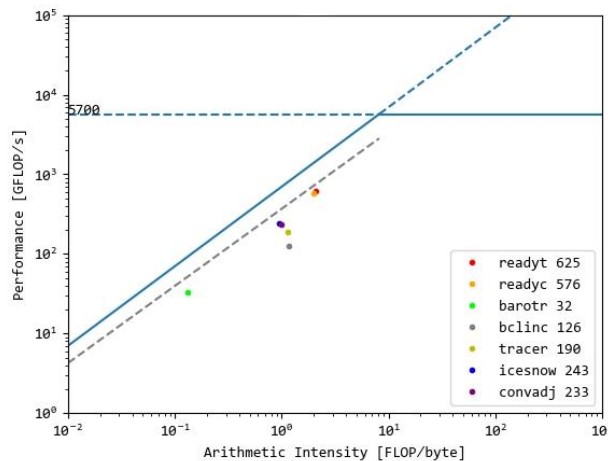

**Figure 9: Roofline model for the AMD GPU and the performance of LICOM's main subroutines.**

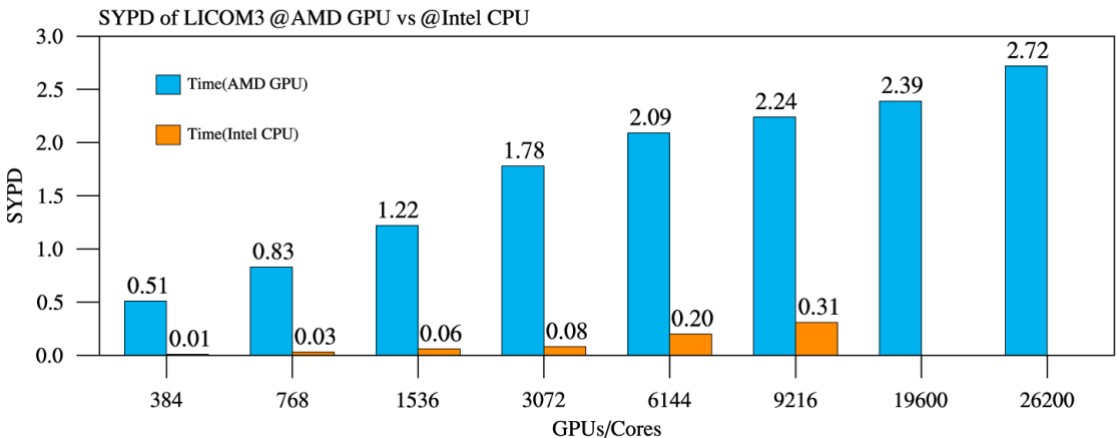

**Figure 10: Simulation performances of the AMD GPU versus Intel CPU core for LICOM3 (1/20°). Unit: SYPD.**

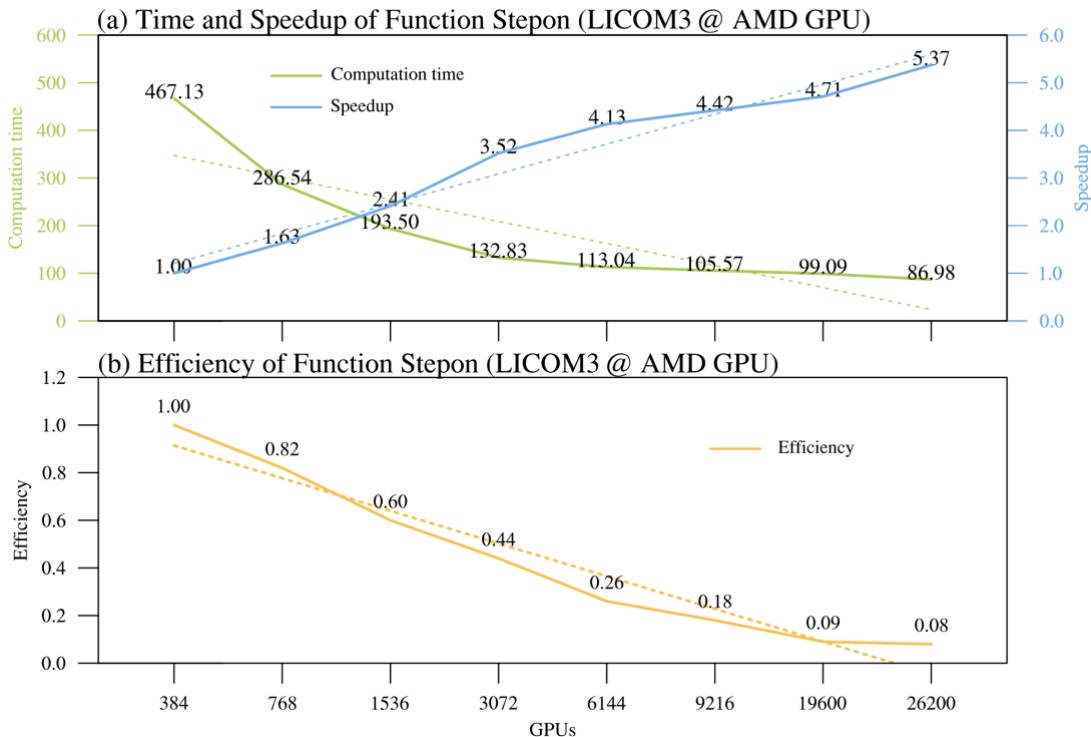


**Figure 11: (a) Computation time (green) and speedup (blue) and (b) parallel efficiency (orange) at different scales for stepons of LICOM3-HIP (1/20°).**

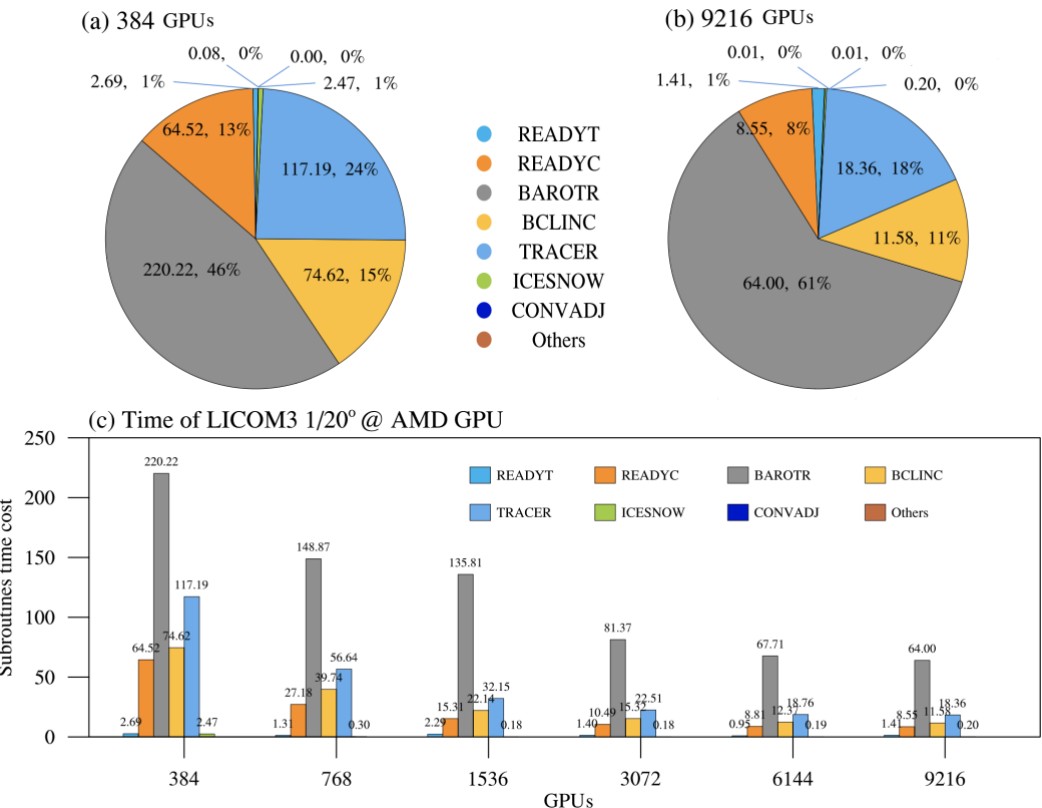

Figure 12: The seven core subroutines' time cost percentages for (a) 384 GPUs and (b) 9216 GPUs. (c) The subroutines' time cost at different scales of LICOM3-HIP (1/20°).

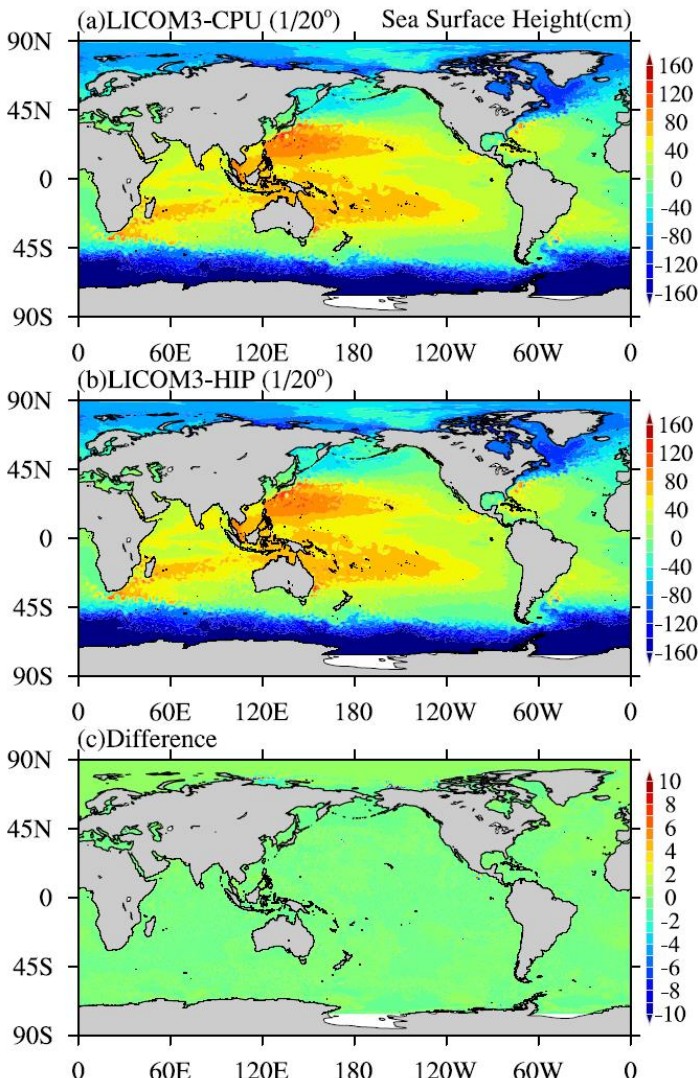

**Figure 13: Daily mean simulated sea surface height for (a) CPU and (b) HIP versions of LICOM3 at 1/20° on March 1st of the 4th model year. (c) The difference between the two versions (HIP minus CPU). Units: cm.**

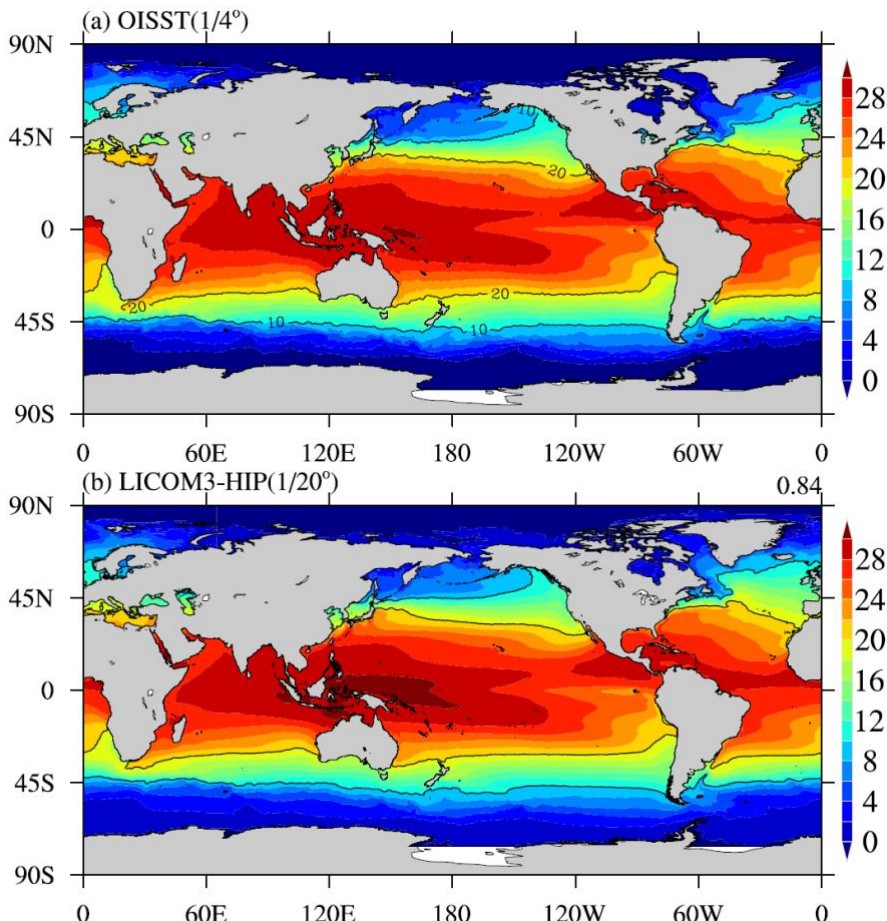

**Figure 14: (a) Observed annual mean sea surface temperature in 2016 from the Optimum Interpolation Sea Surface Temperature (OISST); (b) simulated annual mean SST for LICOM3-HIP at 1/20° during model years 0005-0014. Units: °C.**


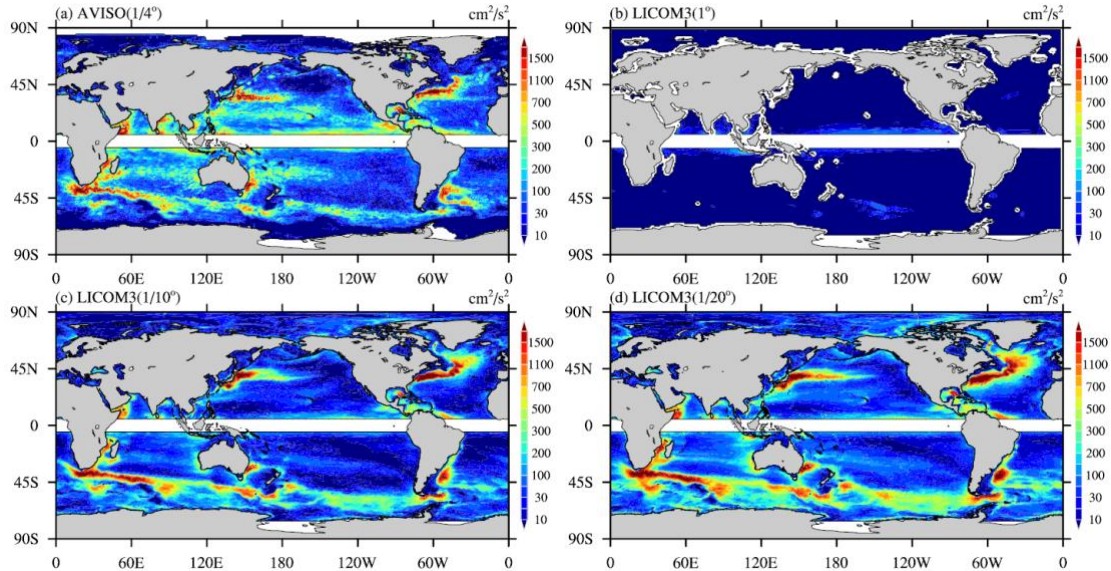

**Figure 15: (a) Observed annual mean eddy kinetic energy (EKE) in 2016 from AVISO. Simulated annual mean SST in 2016 for LICOM3-HIP at (b) 1°, (c) 1/10°, and (d) 1/20°. Units: cm²/s².**


**Table 1: Configurations of the LICOM3 model used in the present study.**

| Experiment | LICOM3-CPU (1°) | LICOM3-HIP(1/10°) | LICOM3-HIP(1/20°) |
|---|---|---|---|
| **Horizontal grid spacing** | 1° (110 km in longitude, approximately 110 km at the equator, and 70 km at mid-latitude) | 1/10° (11 km in longitude, approximately 11 km at the equator, and 7 km at mid-latitude) | 1/20° (5.5 km in longitude, approximately 5.5 km at the equator, and 3 km at mid-latitude) |
| **Gridpoint** | 360×218 | 3600×2302 | 7200×3920 |
| **North Pole** | (65°E, 60.8°N) and (115°W, 60.8°N) | (65°E, 65°N) and (115°W, 65°N) | (65°E, 60.4°N) and (115°W, 60.4°N) |
| **Bathymetry data** | ETOPO2 | Same | Same |
| **Vertical coordinates** | 30 $\eta$ levels | 55 $\eta$ levels | 55 $\eta$ levels |
| **Horizontal viscosity** | Laplacian $A_2$=3000 m$^2$/s | Biharmonic (Fox-Kemper & Menemenlis, 2008) $A_4$=-1.0×10$^9$m$^4$/s | Biharmonic (Fox-Kemper & Menemenlis, 2008)$A_4$=-1.0×10$^8$m$^4$/s |
| **Vertical viscosity** | Background viscosity of 2×10$^{-6}$m$^2$/s with the upper limit of 2×10$^{-2}$m$^2$/s | Background viscosity of 2×10$^{-6}$m$^2$/s with the upper limit of 2×10$^{-2}$m$^2$/s | Background viscosity of 2×10$^{-6}$m$^2$/s with the upper limit of 2×10$^{-2}$m$^2$/s |
| **Time steps** | 120/1440/1440 for barotropic/baroclinic/tracer | 6s/120s/120s for barotropic/baroclinic/tracer | 3s/60s/60s for barotropic/baroclinic/tracer |
| **Bulk Formula** | Large & Yeager (2009) | Same | Same |
| **Forcing data** | JRA55_do, 1958-2018, 6 hourly | JRA55_do, 2016, daily | JRA55_do, 2016, daily |
| **Integration period** | 61 years/6 cycles | 14 years | 14 years |
| **Mixed layer scheme** | Canuto et al. (2001, 2002) | Same | Same |
| **Isopycnal mixing** | Redi (1982); Gent & McWilliams (1990) | Laplacian | Laplacian |

| Bottom drag | $C_b=2.6\times10^{-3}$ | $C_b=2.6\times10^{-3}$ | $C_b=2.6\times10^{-3}$ |
|---|---|---|---|
| **Surface wind-stress** | Relative wind stress | Same | Same |
| **SSS restoring** | 20 m/year; 50 m/30 days for sea ice region | Same | Same |
| **Advection scheme** | Leapfrog for momentum; two-step preserved shape advection scheme for tracer | Same | Same |
| **Time stepping scheme** | Split-explicit Leapfrog with Asselin filter (0.2 for barotropic; 0.43 for baroclinic; 0.43 for tracer) | Same | Same |
| **Sea ice** | Sea ice model of CICE4 | Not coupled | Not coupled |
| **Ref.** | Lin et al. (2020) | This paper | This paper |

**Table 2: Block partition for the 1/20° setup.**

| GPUs | $B_x \times B_y$ | $imt \times jmt$ |
|---|---|---|
| 384 | 600×124 | 604×128 |
| 768 | 600×62 | 604×66 |
| 1536 | 300×62 | 304×66 |
| 3072 | 150×62 | 154×66 |
| 6144 | 100×62 | 104×66 |
| 9216 | 75×62 | 79×66 |
| 19600 | 36×40 | 40×44 |
| 26200 | 36×30 | 40×34 |


**Table 3: The number calls of halos in LICOM3 subroutines for each step.**

| Subroutine | Calls | Calls Percentage |
|---|---|---|
| barotr | 180 | 96.7% |
| bclinc | 2 | 1.1% |
| tracer | 4 | 2.2% |

**Table 4: Some GPU versions of weather/climate models.**

| Model | Language | Max. Grids | Max GPUs | Year and references |
|---|---|---|---|---|
| POM.gpu | CUDA-C | 1922×1442×51 | 4 (K20X) | 2015 (Xu et al., 2015) |
| LICOM2 | OpenACC | 360×218×30 | 4 (K80) | 2019 (Jiang et al., 2019) |
| FUNWAVE | CUDA-Fortran | 3200×2400 | 2 (V100) | 2020 (Yuan et al., 2020) |
| NICAM | OpenACC | 56×56 km×160 | 2560 (K20X) | 2016 (Yashiro et al., 2016) |
| COSMO | OpenACC | 346×340×60 | 4888 (P100) | 2018 (Fuhrer et al., 2018) |
| LICOM3 | HIP | 7200×3920×55 | 26200 (gfx906) | 2020 (This paper) |

**Table 5: Success and failure rates of different scales for one wall clock hour simulation.**

| GPUs | Success | Failure |
|------|---------|---------|
| 384 | 98.85% | 1.15% |
| 1000 | 97.02% | 2.98% |
| 10000 | 72.90% | 27.10% |
| 26200 | 40.19% | 59.81% |