# Peer review of "The GPU version of LICOM3 under the HIP framework and its large-scale application"

_Geoscientific Model Development, 2020_

## Referee Comment (RC1) · Anonymous Referee #1 · 6 Jan 2021

The paper presents implementation of LICOM3 using HIP framework targeting an HPC cluster with AMD GPUs. Atmospheric modeling in not my area of expertise, therefore I will not comment on this aspect of the paper; I will only comment on the model implementation and parallelization efforts. Here are some specific comments about individual sections.

Section 2.3: "The nodes of both partitions are interconnected through the high-performance InfiniBand (IB) networks." Which partitions? Prior system description does not mention any partitions, instead it states that the system consists of 7000 nodes with 4 GPUs per node.

Section 3.1: There is some repetition here, e.g., the authors state several times that HIP supports both AMD and NVIDIA GPUs.

Section 3.2: Good work on converting the original Fortran code to C and verifying the results to be identical between the two implementations. However, the authors do not say much about this ported code in terms of the optimizations applied to it, if the code is purely executed as a single thread on each node, if any 3rd party libraries have been used, etc. Also, how the execution time of this newly ported C version of the code compares to the execution time of the original Fortran code? It is important to have a well-performing CPU code as a starting point.

Section 3.3: From section 3.2, I understood that the entire code was ported to C. However, section 3.3 states that a mixed Fortran/C code was used here. Please clarify to resolve this inconsistency. Also, I do not quite follow the discussion in lines 222-227 about array size. I understand that the original Fortran code uses static arrays and these arrays need to be changed to be dynamically allocated in order to move them to the GPU global memory. Is this what you are implying here? I also have a hard time following the discussion about halo data packing and Fig. 4.

Section 3.4: What storage and file system do you use? These I/O performance numbers are not very meaningful without specifying the underlying storage architecture. It is possible that the low original performance of the I/O operations was solely due to a low-performing storage system and would not be an issue on a higher-performing storage. Also, how exactly is data loaded/stored to disk? The size of data transfers would play a huge role on the overall performance.

Section 4.1: I think the correct way to quantify the speedup is to use all CPU cores vs. the GPUs, not just a single CPU core. Also, it is not clear from the text how the GPUs are managed. For example, is each GPU managed by a single CPU thread, or is the same thread is used to manage all 4 GPUs? Section 5.1: Interesting discussion about failure rates. However, I wonder how realistic your assumptions are with regards to existing studies such as https://www.christian-engelmann.info/publications/gupta17failures.pdf.

Overall, the presented implementation is rather straightforward and not well-thought. The authors ported to GPU several time-consuming kernels within the existing HPC application. Such approach is typically not very productive as it limits the design choices and does not give enough flexibility to optimize the overall application. Before proceeding with such an effort, the authors should have analyzed the amount of time spent in each of the subroutines on the CPU with regards to all other aspects of the code, such as I/O and network communication. Figure 8 shows such kernel time distribution after porting, but I could not find much about the amount of time spent on cross-node communication. It is important to understand the potential benefits of one or another approach before starting the actual implementation effort. Next, there is no discussion about dominant computations in each of the kernels and how to best implement them on the GPU. There is no information in the paper that would indicate anything about the quality of implementation of these GPU kernels, e.g., how well they use GPU resources. I would like to see achieved FLOPS and memory bandwidth of these kernels with respect to the roofline model for this particular GPU to be convinced that the authors did a good job porting these kernels. The discussion about performance improvements is very convoluted by the fact that the authors start comparing performance at some rather large scale of 384 GPUs. What about performance of a single GPU on a much smaller model vs. performance of a single CPU socket, or a 4-GPU node vs. all CPU cores in that node? There is no discussion about what happens inside a single node, for example, how well are all 4 GPUs are utilized and if there is anything that one could benefit from the fact that these 4 GPUs have access to the same host memory. There is no discussion about how the halo exchanges are implemented at the MPI level, how much overlapping is happening for computation, communication, and I/O. The paper is short on many technical details, ranging from characteristics of the underlying hardware, to software/compiler environment, to implementation details, which makes it very difficult for others to put the results obtained by the authors in comparison with other work. Above all, the paper is hard to read due to a very poor use of language; almost every sentence needs corrections.

If the authors want their paper to be printed with the existing implementation, they must provide a much better description and analysis of this implementation.

---

## Referee Comment (RC2) · Anonymous Referee #2 · 15 Feb 2021

Numerical climate modeling is a key method for scientists and researchers to better understand our planet, and one of the most popular applications that greatly challenges the most state-of-the-art high performance computing (HPC) systems. In this work, LICOM3, a standard ocean model is selected and scaled onto the GPU-based heterogeneous supercomputing system. The authors have done lots of porting and optimizing work to put almost all of the time-consuming computation processes into the GPU side, and greatly reduce the communication overhead. As a result, both the dynamic core and the physics part are ported and parallelized on GPUs. A speedup of 42x is achieved when using 284 AMD GPUs VS 384 CPU cores. Excellent scalability is also achieved. A test of 1/20 degree LICOM3-HIP is reached using 6550 nodes and 26200 GPUs, 2.72 SYPD in time-to-solution.

As a computer scientist who also focuses on porting and tuning climate models onto different HPC platforms, dealing with a complete model with lots of code legacies using a new accelerator is obviously not an easy work. Sometimes rewriting and redesigning are necessary to obtain a satisfactory performance. In this work, the optimizing techniques provided are sound and solid, and can be used as a good guidance for corresponding work. AMD GPU and HIP, though not as popular as Nvidia GPU and CUDA for now, are still very promising GPU accelerators for current generation supercomputers. Moreover, it is likely that some of the forthcoming Exa-scale supercomputers, will also be adopting AMD GPUs. So this work is also a good trials ahead of time. More specifically to the strategies: only the most time-consuming parts (seven subroutines) are translated into HIP C, deeply re-coded, ported onto the GPUs, and fully optimized (such as the usage of temporary arrays to avoid data dependency, the change of data structure of original Fortran arrays, etc.). Halos that contain partial communications are handled by CPU part. Therefore, a hybrid computing model is performed, to further improve the overall performance. This is also a very popular strategy when dealing with numerical problems with inter-node or inter-process communications. Besides, The IO part is also considered and tuned by rewriting the data reading strategies and doing parallel scattering.

Overall, the paper is well-structured, with sufficient figures and tables to help better illustrate the ideas where necessary. But there are grammar errors and misleading descriptions here and there. So I suggest the authors ask help from native speakers for further proofreading.

Here are some other suggestions,

The authors mentioned the dynamic and physics parts, but lacks further explanations to what they are. I understand that most communications exist within the dynamic part, but could the authors be more specific in pointing out the optimizing strategies for the dynamic part and the physics part, respectively? Are there any differences?

In the end, the authors claimed that the 1/20° LICOM3-HIP version can not only reproduce the observations, but also produce much smaller scale activities, such as submesoscale eddies and frontal scales structures. Could the authors explain how they obtain the observation version?

Porting a complete model is not an easy work. In this work, approximately 12000 lines Fortran code were rewrote from fortran to C. Could the authors estimate the cost? For example, the number and time cost of persons in the whole project.

The following work is suggested to be cited and comment as well, to enrich the related work part. Optimizing high-resolution Community Earth System Model on a heterogeneous many-core supercomputing platform.

Line 45, I suggest to update the the TOP 500 list using the latest one (Nov. 2020);

Line 49, I don't think the energy result is provided in the work of Xu et al. (2015). Please double check.

Line 71, and the conclusions is in Section 6 –> and the conclusions are in Section 6

Line 74: which started to develop –> which has been developed

Line 83: That makes the coupler is suitable to apply to high resolution modelling. –> It makes the coupler suitable to be applied to high resolution modeling.

Line 85: improve –> improves

Line 103: remove totally

Part 2.3: add some citations or links with more detailed introductions to the supercomputer used in this work.

Line 140: place –> replacing

Line 143: Some... the others... –> Some... some...

Lots of professional words are used in this article, such as theses syntax and macros

used in HIP or CUDA. Please use a different syntax (e.g. italic) for these professional word, to help better identify them. For example, I suspect that 'for tracer', 'baroclinic' and 'barotropic' may refer to professional processes or subroutines in LICOM3, and 'including', 'cuda_', 'hip_', may refer to professional designations of CUDA or HIP framework.

Line 244, an –> a

Figure 5, I suspect the IO time is the result after the IO optimization (part 3.4) being applied. Is that right?

Line 263, times –> time

Please provide more details about the hardware configurations, e.g., the version of CPU and GPUs, the version of complilers, OS, etc.

Please replace Flops/s with Flops.

---

## Author Comment (AC1) · 16 Feb 2021

GMD-2020-323 by Wang et al. Responses to Reviewer(s)

Feb. 10, 2021

I. Response to Reviewer #1
The paper presents implementation of LICOM3 using HIP framework targeting an HPC cluster with AMD GPUs. Atmospheric modeling in not my area of expertise, therefore I will not comment on this aspect of the paper; I will only comment on the model implementation and parallelization efforts. Here are some specific comments about individual sections.

Response: Thanks for your insightful comments. We were planning to document the final version and test at first. Therefore, we only present the final version and the results from a large-scale test in the manuscript. Another paper shows details of the model porting and results of the small-scale test are preparing. Based on your suggestions, we have done some additional tests, added much more information of the system's software and hardware, and further polished this manuscript's English. Also, we have modified the format of the references. The point-to-point responses are listed below.

Section 2.3: "The nodes of both partitions are interconnected through the high performance InfiniBand (IB) networks." Which partitions? Prior system description does not mention any partitions, instead it states that the system consists of 7000 nodes with 4 GPUs per node.

Response: Thanks for your questions. In fact, the implementation of the network is a 3-level fat-tree architecture using Mellanox 200Gb/s HDR InfiniBand, whose measured point-to-point communication performance is about 23GB/s. The supercomputer consists of 6 partitions (or rings, each has 1200 nodes, and the job system schedules about 7200 nodes). We have modified the descriptions in the revised manuscript in Section 2.3, Lines 132-136.

Section 3.1: There is some repetition here, e.g., the authors state several times that HIP supports both AMD and NVIDIA GPUs.

Response: Thanks. We have revised the repetition in Section 3.1.

Section 3.2: Good work on converting the original Fortran code to C and verifying the results to be identical between the two implementations. However, the authors do not say much about this ported code in terms of the optimizations applied to it, if the code

is purely executed as a single thread on each node, if any 3rd party libraries have been used, etc. Also, how the execution time of this newly ported C version of the code compares to the execution time of the original Fortran code? It is important to have a well-performing CPU code as a starting point.

Response: Thanks for your comments. First, we would like to correct the statement of this porting. Actually, we have only ported the main computation part of LICOM3 from Fortran code to C, which includes the seven kernels listed in Figure 2 (readyt, readyc, barotr, bclinc, tracer, icesnow, and convadj). All seven kernels account for about 25% of the code and about 99% of the computation and communication time (Table R1). Other parts, mainly the initialization, the time integration control, and the IO parts, still use Fortran code. Therefore, the so-called C version of LICOM3 is a Fortran and C mixed code, not a pure C code. In this way, we can replace F90 subroutines one by one and easily check the correctness of ported C codes. The C code is purely executed as a single thread on each node.

The Fortran and C mixed code is the starting point of following CUDA-C or HIP programming. To guarantee the correctness of the porting, we did NOT optimize the model in this step, and there were not any additional 3rd party libraries used. We simply translated the Fortran code to C code sentence by sentence. The translations include converting array index from Fortran (start from 1) to C (start from 0), changing the array subscript sequence, etc. Since the code is the same for 100km and 5km, we only test the Fortran and C mixed version for lower resolution cases.

We have tested the byte-to-byte correctness of the Fortran and C mixed code and tested the execution time. The Fortran and C hybrid version's speed is slightly faster than the original Fortran code, less than 10%. Figure R1 shows a speed benchmark by the LICOM3 for 100km and 10km running on an Intel platform. The results are the time running one model month for a low-resolution test and one model day for a high-resolution test. The details of the platform are in the caption of Figure R1. This indicates that we have successfully ported these kernels from Fortran to C.

We have added more descriptions of the porting and added a figure (Figure R1) in the revised manuscript in Section 3.2, Lines 187-195.

Section 3.3: From section 3.2, I understood that the entire code was ported to C. However, section 3.3 states that a mixed Fortran/C code was used here. Please clarify to resolve this inconsistency. Also, I do not quite follow the discussion in lines 222-227 about array size. I understand that the original Fortran code uses static arrays and these arrays need to be changed to be dynamically allocated in order to move them to the GPU global memory. Is this what you are implying here? I also have a hard time following the discussion about halo data packing and Fig. 4.

Response: Thanks for your questions. Sorry for the confusing statements. As our response to the previous section, the code of LICOM3 is not a pure C code but a Fortran and C mixed one. We have modified the vague statement in the manuscript to avoid this inconsistency in Lines 187-189.

The "lines 222-227" problem is not really what you understand. It occurs for the 5km resolution version when the static data array beyond 2GB. We met this problem in both the HIP hipcc and CUDA nvcc compiler. It is perhaps due to the compiler limitation of the GPU compilation tool. We have revised this part to state the problem clearly, in Lines 228-230.

To optimize the heavy halo in the kernel "barotr", we have tried to reduce the amount of data using the halo data packing method. Only the necessary data are transferred. The test indicates that the method can decrease by about 30% of the total wall clock time of "barotr" when 384 GPUs used. But we have not optimized other kernels so far because its performance not as good as 384GPUs' when GPUs scale beyond 10000. This optimization is not turn on in the following experiments, and we just keep it here as an option to improve the performance of 'barotr'. Other methods, such as NCCL (NVIDIA Collective Communication Library), are also implemented and tested, but not presented here. We have revised this part to state the problem clearly, in Lines 252-

255.

Section 3.4: What storage and file system do you use? These I/O performance numbers are not very meaningful without specifying the underlying storage architecture. It is possible that the low original performance of the I/O operations was solely due to a low-performing storage system and would not be an issue on a higher-performing storage. Also, how exactly is data loaded/stored to disk? The size of data transfers would play a huge role on the overall performance.

Response: Thanks. The storage file system of the supercomputer is ParaStor300S with a 'parastor' file system, whose measured write and read performance is about 520GB/s and 540 GB/s. We have added this information in the revised manuscript, Lines 138-139.

We agree with you that the I/O time depends on system I/O performance. When the large scale LICOM3 tests are performing, the system is forced to be one user mode by the administrator. It means no other users' job and heavy I/O tasks are running at the same time. Hence, the I/O of storage is enough to hold all LICOM3's I/O tasks. The improvement of I/O performance has been depicted in Figure 5 of the manuscript. The ratio of I/O in the total calculation time is less than 50% for 9216 cards, which is almost 90% before the I/O processes have been optimized.

After optimization, the forcing data, about 3GB total, are read from disk every model year, while the daily mean predicted variables, about 60GB total, are stored to disk every model day. Therefore, the performance of output dominated the I/O performance. We have added the above statements in the revised manuscript, Lines 256-257.

Section 4.1: I think the correct way to quantify the speedup is to use all CPU cores vs. the GPUs, not just a single CPU core. Also, it is not clear from the text how the GPUs are managed. For example, is each GPU managed by a single CPU thread, or is the same thread is used to manage all 4 GPUs?

Response: Thanks. We agree with you and have already computed the speedup using the one CPU, but not shown in the manuscript. Since each node has 32 CPU cores and 4 GPUs, each GPU is managed by one CPU thread in the present cases. We can also quantify GPUs' speedup vs. all CPU cores' on the same number of nodes. For example, the 384 (768) GPUs correspond to 96 (192) nodes, which have 3072 (6144) CPU cores. Therefore, the overall speedup is about 6.375 (0.51/0.08) for 384 GPUs and 4.15 (0.83/0.2) for 768 GPUs (Figure 6). The speedups are comparable with our previous work porting LICOM2 to GPU using OpenACC (Jiang et al., 2019), which is about 1.8-4.6 times speedup using one GPU card vs. two 8-core Intel GPU in small-scale experiments for specific kernels. Our results are also slightly better than Xu et al. (2015), which has ported another ocean model to GPU using Cuda C. But due to the limitation of the number of intel CPUs (maximal 9216 cores), we didn't obtain the overall speedup for 1536 and more GPUs. We have added discussions about this issue in the revised manuscript, Lines 300-307.

Section 5.1: Interesting discussion about failure rates. However, I wonder how realistic your assumptions are with regards to existing studies such as https://www.christianengelmann.info/publications/gupta17failures.pdf.

Response: Thanks for your suggestion and the interesting paper. We have read this manuscript and have cited it in the revised version. Unlike Gupta's study (23 types of failures), only three types of failures we have mostly met are discussed in Section 5.1. We only do simple analysis by predefined hypothesis (failure occur rateãĂ"10" ãĂˆˆ"-5" , which means 1 failure in 100000 hours.) to illustrate the difficulty of submitting jobs beyond 10000 GPUs. The realistic failures are essential, but not the critical problem of this study, so we did not discuss it more. Lines 360-364.

Overall, the presented implementation is rather straightforward and not well-thought. The authors ported to GPU several time-consuming kernels within the existing HPC application. Such approach is typically not very productive as it limits the design choices and does not give enough flexibility to optimize the overall application. Before proceeding with such an effort, the authors should have analyzed the amount of time spent in each of the subroutines on the CPU with regards to all other aspects of the code, such as I/O and network communication. Figure 8 shows such kernel time distribution after porting, but I could not find much about the amount of time spent on cross-node communication. It is important to understand the potential benefits of one or another approach before starting the actual implementation effort.

Response: Thanks for your comments and suggestions. Before the implementation, we actually have tested the performances of all seven Fortran subroutines on a super-computer system using Intel CPUs. Figure R2 shows our test results, including the seven subroutines' time cost percentage for 384 and 9216 CPU cores and the subroutines' time cost at six scales. Here, we used the 1/20° resolution LICOM3. Although the two experiments have been conducted on different platforms, the two model versions' configurations are identical, particularly using the same time steps, 1s for the barotropic part, the 60s for both baroclinic and tracer parts. That will grantee the same numbers of halo for two versions. Because the code of I/O in Fortran and the Fortran-C mixed version is the same, we will not discuss more I/O for the original Fortran code.

We found that the "tracer" is the most time-consuming subroutine for the CPU version. With the increase of CPU cores from 384 to 9216, the ratio of cost time for "tracer" is also increasing from 38% to 49%. "readyt" and "readyc" are computing intensive subroutines. "tracer" is both computing intensive and communication intensive subroutine. "barotr" is communication intensive subroutine and the communication of "barotr" is 45 times than that of "tracer". The computing intensive subroutines can achieve good speedup by GPU, but the communication intensive subroutine will achieve poor performance.

We have done a set of experiments to measure the time cost of both halo update and memory copy in the HIP version. These two processes in the time integration are conducted in three subroutines: "barotr", "bclinc," and "tracer". The figure shows that "barotr" is the most time-consuming subroutine, and the memory copy dominates,

which takes about 40% of the total time cost. These results can also be treated as a reference for the CPU experiments because the Halo update codes in the HIP version are the same as the CPU version.

We have added discussions about the above issues in the revised manuscript, Lines 240-254.

Next, there is no discussion about dominant computations in each of the kernels and how to best implement them on the GPU. There is no information in the paper that would indicate anything about the quality of implementation of these GPU kernels, e.g., how well they use GPU resources. I would like to see achieved FLOPS and memory bandwidth of these kernels with respect to the roofline model for this particular GPU to be convinced that the authors did a good job porting these kernels. The discussion about performance improvements is very convoluted by the fact that the authors start comparing performance at some rather large scale of 384 GPUs. What about performance of a single GPU on a much smaller model vs. performance of a single CPU socket, or a 4-GPU node vs. all CPU cores in that node? There is no discussion about what happens inside a single node, for example, how well are all 4 GPUs are utilized and if there is anything that one could benefit from the fact that these 4 GPUs have access to the same host memory.

Response: Thank you very much for your suggestion. This paper's critical point is developing a high-resolution heterogeneous version of the ocean circulation model, so we just give out some brief introduction of GPU implementation and optimization methods in section 3. Follow the Reviewer's suggestions; we append more discussions about floating-point operations performance of the subroutines in section 4.1. The addition roofline test is archived by a 100km version.

Figure R4 shows the roof-line model using the Stream-GPU and LICOM program's measured behavioral data on a single computation node bound to one GPU Card depicting the relationship between arithmetic intensity and performance floating points

operations.

The blue and gray oblique line is the fitting line related to the Stream-GPU program's behavioral data using 5.12e8 and 1e6 threads, respectively, both with blocksize of 256, which attain the best configuration. For details, the former is approximately the maximum threads number restricted by GPU card memory achieving the bandwidth limit to 696.52 GB/s. In comparison, the latter is close to the average number of threads in GPU parallel calculations used by LICOM, reaching the bandwidth of 344.87 GB/s on average. Here we use the oblique gray line as a benchmark to verify the rationality of LICOM's performance, accomplishing the bandwidth of 313.95 GB/s averagely.

Due to the large calculation scale of the whole LICOM program, the divided calculation grid bound to a single GPU card is limited by video memory; most kernel functions actually issue no more than 1.2e6 threads. As a result, the floating-point operations performance is a little far from the oblique roof-line shown in Figure R3. In particular, the subroutine bclinc apparently stray off the whole trend for the reason of including frequent 3d-array Halo MPI communications as well as a lot of data transmission between CPU and GPU.

There is no discussion about how the halo exchanges are implemented at the MPI level, how much overlapping is happening for computation, communication, and I/O.

Response: Thanks for your suggestion. The halo exchange in the MPI level is similar to POP have (see Jiang, et,al. 2019). We did not change these codes in the HIP version. We call an F90 subroutine to do halo from GPU space. The overlapping between communication data packing/unpacking and point-to-point communication was implemented. We did not apply the overlapping of computation, communications, and I/O in this version because the calculation time was much less than communication time and I/O time. There have no existing solutions to increase these halo performances; we hope it can be improved in the future LICOM3 version. This work's core contribution is to develop a 10000+ GPUs runnable ocean model, which still has sped up in

26200 scales. The seven core computation process put into GPUs space by the HIP framework is the critical solution. All other are not important issues because they cost 99%+ computation time in the model 'stepon' procedure. The brief hardware and software environments are presented in the manuscript. Since all supercomputer has their unique settings, this model may have different performance in other computers than we have.

The paper is short on many technical details, ranging from characteristics of the underlying hardware, to software/compiler environment, to implementation details, making it very difficult for others to put the results obtained by the authors compared to other work. Response: Thanks. We have revised our manuscript following your suggestions, and We have tried to provide much more information as we can.

Above all, the paper is hard to read due to a very poor use of language; almost every sentence needs corrections.

Response: Thanks for your suggestion. We have revised the manuscript as possible and will invite a 3rd part English editor to improve the revised manuscript.

If the authors want their paper to be printed with the existing implementation, they must provide a much better description and analysis of this implementation. Interactive comment on Geosci. Model Dev. Discuss., https://doi.org/10.5194/gmd-2020-323, 2020.

Response: Thanks. We have revised our manuscript following your suggestions, and We have tried to provide much more information as we can.

Reference

Jiang, J. R., P. F. Lin, J. Wang, H. L. Liu, X. B. Chi, H. Q. Hao, Y. Z. Wang, W. Wang, and L. H. Zhang, 2019: Porting LASG/IAP Climate system Ocean Model to GPUs using OpenACC. IEEE Access, 7(1), 154490-154501. doi:10.1109/ACCESS.2019.2932443

Xu, S., Huang, X., Oey, L. Y., Xu, F., Fu, H., Zhang, Y., and Yang, G. (2015). POM. gpu-v1. 0: a GPU-based Princeton Ocean Model. Geoscientific Model Development,

8(9), 2815-2827. https://doi.org/10.5194/gmd-8-2815-2015

Table R1 The seven major subroutines' time cost for one model day at different scales for LICOM3 (1/20°). The time cost of I/O and total time are also listed. These tests were conducted on a platform of Intel Xeon CPU (E5-2697A v4, 2.60GHz). Unit: second.

|  | 384 cores | 768 cores | 1536 cores | 3072 cores | 6144 cores | 9216 cores |
|---|---|---|---|---|---|---|
| readyt | 1678.93 | 784.14 | 454.38 | 278.18 | 142.23 | 95.28 |
| readyc | 4751.21 | 2244.72 | 939.51 | 463.71 | 228.88 | 146.45 |
| barotr | 3449.31 | 1628.74 | 642.58 | 396.34 | 143.41 | 91.86 |
| bclinc | 1431.89 | 696.73 | 363.93 | 234.59 | 137.11 | 101.76 |
| tracer | 7439.98 | 3147.73 | 1822.47 | 1467.40 | 614.94 | 445.63 |
| icesnow | 31.54 | 17.17 | 8.61 | 4.80 | 2.85 | 2.37 |
| convadj | 620.61 | 370.03 | 193.95 | 90.87 | 42.50 | 29.56 |
| I/O | 150.52 | 128.62 | 134.42 | 132.80 | 152.85 | 157.15 |
| Total | 19553.98 | 9017.89 | 4559.86 | 3068.69 | 1464.76 | 1070.06 |

Figure Captions:

Figure R1. The wall clock time of a model day for the 10km version and a model month for the 100km version. The blue and orange bars are for the Fortran and the Fortran and C mixed version. These tests were conducted on a platform of Intel Xeon CPU (E5-2697A v4, 2.60GHz). We used 28 and 280 cores for the low and high resolution, respectively.

Figure R2 The seven core subroutines' time cost percentage for (a) 384 and (b) 9216 CPU cores. (c) the subroutines' time cost at different scales of LICOM3 (1/20°). These tests were conducted on a platform of Intel Xeon CPU (E5-2697A v4, 2.60GHz).

Figure R3 The ratio of the time cost of halo update and memory copy to the total time cost for three subroutines, "barotr" (green), "bclinc" (blue), and "tracer" (orange) in the HIP version LICOM for three scales (Unit: %). The numbers in the blankets are the time cost of the two processes (Unit: second).

Figure R4 Roofline model for AMD GPU and the performance of LICOM's main subroutines.

[Figure]

**Fig. 1.** The wall clock time of a model day for the 10km version and a model month for the 100km version. The blue and orange bars are for the Fortran and the Fortran and C mixed version.

[Figure]

**Fig. 2.** The seven core subroutines' time cost percentage for (a) 384 and (b) 9216 CPU cores. (c) the subroutines' time cost at different scales of LICOM3 (1/20°).

[Figure]

**Fig. 3.** The ratio of the time cost of halo update and memory copy to the total time cost for three subroutines, "barotr" (green), "bclinc" (blue), and "tracer" (orange) in the HIP version.

[Figure]

**Fig. 4.** Roofline model for AMD GPU and the performance of LICOM's main subroutines.

---

## Author Comment (AC2) · 26 Feb 2021

Numerical climate modeling is a key method for scientists and researchers to better understand our planet, and one of the most popular applications that greatly challenges the most state-of-the-art high performance computing (HPC) systems. In this work, LICOM3, a standard ocean model is selected and scaled onto the GPU-based heterogeneous supercomputing system. The authors have done lots of porting and optimizing work to put almost all of the time-consuming computation processes into the GPU side, and greatly reduce the communication overhead. As a result, both the dynamic core and the physics part are ported and parallelized on GPUs. A speedup of 42x is achieved when using 284 AMD GPUs VS 384 CPU cores. Excellent scalability is also achieved. A test of 1/20 degree LICOM3-HIP is reached using 6550 nodes and

26200 GPUs, 2.72 SYPD in time-to-solution.

As a computer scientist who also focuses on porting and tuning climate models onto different HPC platforms, dealing with a complete model with lots of code legacies using a new accelerator is obviously not an easy work. Sometimes rewriting and redesigning are necessary to obtain a satisfactory performance. In this work, the optimizing techniques provided are sound and solid, and can be used as a good guidance for corresponding work. AMD GPU and HIP, though not as popular as Nvidia GPU and CUDA for now, are still very promising GPU accelerators for current generation supercomputers. Moreover, it is likely that some of the forthcoming Exa-scale supercomputers, will also be adopting AMD GPUs. So this work is also a good trials ahead of time. More specifically to the strategies: only the most time-consuming parts (seven subroutines) are translated into HIP C, deeply re-coded, ported onto the GPUs, and fully optimized (such as the usage of temporary arrays to avoid data dependency, the change of data structure of original Fortran arrays, etc.). Halos that contain partial communications are handled by CPU part. Therefore, a hybrid computing model is performed, to further improve the overall performance. This is also a very popular strategy when dealing with numerical problems with inter-node or inter-process communications. Besides, The IO part is also considered and tuned by rewriting the data reading strategies and doing parallel scattering.

Overall, the paper is well-structured, with sufficient figures and tables to help better illustrate the ideas where necessary. But there are grammar errors and misleading descriptions here and there. So I suggest the authors ask help from native speakers for further proofreading.

Response: Thank you very much for your comments and suggestions. We have revised the English of the manuscript as possible and will find a professional English editor to further improve the final manuscript.

Here are some other suggestions, The authors mentioned the dynamic and physics

parts, but lacks further explanations to what they are. I understand that most communications exist within the dynamic part, but could the authors be more specific in pointing out the optimizing strategies for the dynamic part and the physics part, respectively? Are there any differences?

Response: Thanks. To explicitly separate the dynamic core and the physical package is an excellent ideal for further optimization. But so far, the optimizing strategies are mostly at the program level, not treat the dynamic or physics parts separately. We only ported all seven core subroutines within the time integration loops to GPU, including both the dynamic and physics parts.

Unlike the atmospheric models, there are no many time-consuming physical processes in the ocean model, such as the radiative transportation, cloud, precipitation, and convection processes. Therefore, the two kinds of parts are usually not clearly separated in the ocean model, particular in the early stage of model development. This is also the case of LICOM. We have added the discussion of this issue in the revised version, Lines 440-445.

In the end, the authors claimed that the 1/20_ LICOM3-HIP version can not only reproduce the observations, but also produce much smaller scale activities, such as submesoscale eddies and frontal scales structures. Could the authors explain how they obtain the observation version?

Response: Thanks. This is kind of misleading. So far, the horizontal resolution of most global-scale observation is commonly no more than 25km from merged remote sensing products, which cannot resolve the submesoscale eddies in most places of the ocean. Some products, such as sea surface temperature, indeed have higher resolution at several kilometers. But these products usually have either short period or limited region, and not suitable for the global-scale, long-term climate research. Here, we only would like to say that much finer scale processes can be captured in this 1/20 model and didn't intend to compare with the fine scale observation.

We have revised the sentences and tied to avoid misunderstanding, in Lines 424-425.

Porting a complete model is not an easy work. In this work, approximately 12000 lines Fortran code were rewrote from fortran to C. Could the authors estimate the cost? For example, the number and time cost of persons in the whole project.

Response: Thanks. This is a good question. To port the Fortran code to C costs us about five months (2018.11-2019.3), and five Ph.D. students and five part-time staff participated in this programming work. Then, it took about ten months to port these C codes to GPU in CUDA (2019.4-2020.1) and further four months (2020.2-2020.5) to optimize them on the HIP framework, such as introducing IO parallel and doing the large-scale test. Therefore, it totally took nineteen months, and five Ph.D. students and five part-time staff to finish this kind of porting work. We have added the discussion of this issue in the revised version, Lines 406-407.

The following work is suggested to be cited and comment as well, to enrich the related work part. Optimizing high-resolution Community Earth System Model on a heterogeneous many-core supercomputing platform.

Response: Thanks for your suggestion and for providing a new reference. We have cited it in the revised paper in Lines 55-57.

Line 45, I suggest to update the TOP 500 list using the latest one (Nov. 2020);

Response: Thanks. We updated the list in the revised manuscript, Lines 45-47.

Line 49, I don't think the energy result is provided in the work of Xu et al. (2015). Please double check.

Response: Thanks. We have checked the work of Xu et al. (2015). They did provide the energy result in Sub-section 5.3.4 of their paper, shown in the following image (Figure R1).

Line 71, and the conclusions is in Section 6 –> and the conclusions are in Section 6
Line 74: which started to develop –> which has been developed Line 83: That makes the coupler is suitable to apply to high resolution modelling. –> It makes the coupler suitable to be applied to high resolution modeling. Line 85: improve –> improves Line 103: remove totally

Response: Thanks for your careful reading. We have corrected all five typos in the revised manuscript.

Part 2.3: add some citations or links with more detailed introductions to the supercomputer used in this work.

Response: Thanks. Because there is no publication about this supercomputer, we have added some information about this machine, including InfiniBand network speed and structure, the speed of the storage file system, etc. We have added this information in the revised manuscript, please see Section 2.3.

Line 140: place –> replacing Line 143: Some... the others... –> Some... some...

Response: Thanks for your careful reading. We have corrected these two typos in the revised manuscript.

Lots of professional words are used in this article, such as theses syntax and macros used in HIP or CUDA. Please use a different syntax (e.g. italic) for these professional word, to help better identify them. For example, I suspect that 'for tracer', 'baroclinic' and 'barotropic' may refer to professional processes or subroutines in LICOM3, and 'including', 'cuda_', 'hip_', may refer to professional designations of CUDA or HIP framework.

Response: Thanks for your suggestion. We have revised the manuscript to avoid misunderstanding. Now "barotr", "HipMemcpy" and etc., which inside "" are the function name in .cpp source file. The "cuda" and "hip" in Lines 154 is the prefix for the conversion of function call (or header files) from CUDA style to HIP style. For example "CudaMemcpy" need to be changed to "HipMemcpy", and "cuda_runtime.h" to

"hip_runtime.h".

Line 244, an –> a

Response: Thanks. Corrected.

Figure 5, I suspect the IO time is the result after the IO optimization (part 3.4) being applied. Is that right?

Response: Thanks. Yes, it is correct.

Line 263, times –> time

Response: Thanks. Corrected.

Please provide more details about the hardware configurations, e.g., the version of CPU and GPUs, the version of compilers, OS, etc.

Response: Thanks. We have added all the information in the revised manuscript following your suggestions in Section 2.3.

Please replace Flops/s with Flops.

Response: Thanks. Replaced.
* * *
two GPUs and 92 % on four GPUs. When more GPUs are used, the size of each subdomain becomes smaller. This decreases the performance of POM.gpu in two aspects. First, the communication overhead may exceed the computation time of the inner region as the size of each subdomain decreases. As a result, the overlapping methods in Sect. 4.2 are not effective. Second, there are many "small" kernels in the POM.gpu code, in which the calculation is simple and less time-consuming. With fewer inner region computations, the overhead of kernel launching and implicit synchronization with kernel execution must be counted.

**5.3.4 Comparison with a cluster**

In the last test, we compare the performance of POM.gpu on a workstation containing four GPUs with that on the $Tansuo100$ cluster. Three different high-resolution grids (Grid-1: $962 \times 722 \times 51$; Grid-2: $1922 \times 722 \times 51$; Grid-3: $1922 \times 1442 \times 51$) are used. Figure 13 shows that our workstation with four GPUs is comparable to 408 standard CPU cores (= 34 nodes $\times$ 12 cores/node) in the simulation. Because the thermal design power of one X5670 CPU is 95 W and that of one K20X GPU is 235 W, we reduce the energy consumption by a factor of 6.8. Theoretically, as the subdo-

Geosci. Model Dev., 8, 2815–2827, 2015

www.geosci-model-dev.net/8/2815/2015/

**Fig. 1.** Image from Xu et al. (2015).

---

## Referee Report (RR1)

The paper describes the GPU parallelization of LICOM3, good parallel speed ups and good scalability toward a large number of GPUs has been obtained. The paper focusses on GPU programming and code optimization rather than model development (in this case scope ocean model). However, given the scope of the journal, I would advise to include some introduction of components/program modules of LICOM3 in terms of ocean dynamics. Furthermore, there are too many inaccurate or incorrect language usage, the paper needs to be corrected by a native English speaker.

Detailed comments/questions:

Figure 5, Super linear speed ups are observed for some tracer and readyc from 384 to 768 GPUs. What are the reasons?

Section 6. Conclusions contains many detailed discussions, the detailed discussions should be in 5. Discussion, and only briefly summarize the main conclusions in 6.

Line 16, explain "3-dimensional parallelization"
Line 69, model's best computing performance? Do you mean only the best or fastest results are reported?
Line 160, What are "GPU space nodes"?
Line 169, What is "Ocean block distribution"? Describe.
Line 170, The description is too much in terms of code execution steps (loops). Describe tracer, barocline and barotropic in terms of sub-models of ocean dynamics.
Line 179," The results are the time running one model month", do you mean: The time reported is the wall clock time of running one model month?
Line 214, "the 3-D parallelism is implemented", what is 3-D parallelism? please elaborate
Line 259, "equivalent to one step", one step of what?
Line 261, "we then rewrite the data reading strategy and do parallel scattering for ten different forcing variables", needs more explanation.
Line 263, What is "the core-process"?
Line 389, What are "The dynamic core and physic packages"?

Some minor corrections:

Line 20, Change "can still obtain an increasing," to "can still be increased to"
Line 60, Change "is supported by" to "provides support for"
Line 81, Change "preparing" to "in preparation"
Line 109, Change "totally" to "in total,"
Line 111, grammar
Line 112, Change "are suitable for" to "require" or "demand"
Line 154, Remove "including"
Line 164, Change "earlier" to "ago"
Line 196, Change "grids were united " to "grid points are grouped"
Line 217, "parallel" to "parallelize"

Line 223, "parallel" to "parallelization"
Line 245, "Variable" to "Floating point" or "Arithmetic"
Line 259, "updated" to "increased"
Line 269, Change "0.2-0.3 SYPD will require too much" to "at a speed of 0.2-0.3 SYPD it will take too long"

---

## Author Response (AR2)

The paper describes the GPU parallelization of LICOM3, good parallel speed ups and good scalability toward a large number of GPUs has been obtained. The paper focusses on GPU programming and code optimization rather than model development (in this case scope ocean model). However, given the scope of the journal, I would advise to include some introduction of components/program modules of LICOM3 in terms of ocean dynamics. Furthermore, there are too many inaccurate or incorrect language usage, the paper needs to be corrected by a native English speaker.

**Response:** Thanks for your comments and suggestions. Following your suggestions, we have added some descriptions in Section 2.1 to introduce the components and program modules of LICOM3 briefly as follows, particularly for the seven subroutines porting to GPU. We have further revised the English of the manuscript and have ask a professional English editor to polish it. The editing certification is included at the end of this letter. The point-to-point responses are listed as follows.

"*The essential task of the ocean model is to solve the approximated Navier-Stocks equations, along with the conservation equations of the temperature and salinity. Seven kernels are within the time integral loop, named "readyt", "readyc", "barotr", "bclinc", "tracer", "icesnow", and "convadj", which are also the main subroutines porting from the CPU to the GPU. The former two kernels computed the terms in the barotropic and baroclinic equations of the model. The following three ("barotr", "bclinc", and "tracer") are used to solve the barotropic, baroclinic, and temperature/salinity equations. The last*

*two subroutines deal with the seaice and the deep convection processes at the high latitudes. All these subroutines have about 12000 lines of source code, accounting for approximately 25% of the total code and 95% of computation.*"

Detailed comments/questions:

35 Figure 5, Super linear speed ups are observed for some tracer and readyc from 384 to 768 GPUs. What are the reasons?

**Response:** Thanks. The speed of the model (also individual subroutines) not only depends on the number of cores but also on the speed of communication or the usage of memory/cache, particularly for the stencil problem in the present study. Based on our analysis, the kernels' (tracer and readyc) superlinear speedups are mainly caused by memory usage. That is, the memory

40 usage of each thread for 768 GPU cards is only half for 384 GPU cards. That may cause the superlinear speedups.

Section 6. Conclusions contains many detailed discussions, the detailed discussions should be in 5. Discussion, and only briefly summarize the main conclusions in 6.

45 **Response:** Thank you very much for your suggestion. This is also what we had thought about when we wrote the paper. Because the discussions in Section 6 were a little scattered and seem to be closely related to the conclusions, we didn't put these discussions in Section 5. Also, further development of the model is discussed here. To make the manuscript much clearly, we do not move these discussions.

50 Line 16, explain "3-dimensional parallelization"

**Response:** Thanks. When solving the ocean circulation equations numerically, the seawater is used to discrete to a 3-dimensional grid (x, y and z). Usually, the grid has been horizontally partitioned into latitude belts (y direction) or longitude-latitude boxes (x and y direction) for parallelization. Here, "3-dimensional parallelization" means the grid has also been

55 partitioned in the depth (or z direction) direction. We have added the explanation in the revised manuscript.

Line 69, model's best computing performance? Do you mean only the best or fastest results are reported?

**Response:** Yes. Only the best or fastest results are reported. Because several other users were also conducting tests when we

60 run ours. Therefore, the results may be affected by the other jobs. We think that the best or fastest results may reflect the actual ability of the machine.

Line 160, What are "GPU space nodes"?

65    **Response:** Thanks. "GPU space nodes" are confusing. We actually want to say "GPU memory spaces at different nodes". We modified it in the revised manuscript.

Line 169, What is "Ocean block distribution"? Describe.

70    **Response:** Thanks. "Ocean block distribution" means to distribute the data on the partitioned grid to each thread. We revised this expression in the manuscript.

Line 170, The description is too much in terms of code execution steps (loops). Describe tracer, barocline and barotropic in terms of sub-models of ocean dynamics.

75

    **Response:** Thanks. We have been added brief descriptions of seven kernels in the revised manuscript in Section 2.1. You also suggested this at the beginning of your comments.

Line 179," The results are the time running one model month", do you mean: The time reported is the wall clock time of
80    running one model month?

    **Response:** Thanks. Yes, it is. We changed it to "The results are the wall clock time of running one model month".

Line 214, "the 3-D parallelism is implemented", what is 3-D parallelism? please elaborate

85

    **Response:** Thanks. "the 3-D parallelism" is actually "the 3-dimensional parallelism". When solving the ocean circulation equations numerically, the seawater is used to discrete to a 3-dimensional grid (x, y, and z). Usually, the grid has been horizontally partitioned into latitude (or y) belts or longitude-latitude (or x-y) boxes for parallelization on CPU or GPU. Here, "3-dimensional parallelization" means the grid has also been partitioned in the depth (or z) direction. Initially, the LICOM3
90    are using only 2-D MPI parallelism in the x and y directions. To increase the scalability of the HIP version, we also partitioned the grid in the z-direction. We further explained this term in the revised manuscript.

Line 259, "equivalent to one step", one step of what?

95    **Response:** Thanks. We want to express that the data reading time is comparable to the wall clock time for one model step. We modified it in the revised manuscript.

Line 261, "we then rewrite the data reading strategy and do parallel scattering for ten different forcing variables", needs more explanation.

100

**Response:** Thanks. We decreased the reading frequency from every day to every month but using large arrays for the input. Originally, 10 variables are sequentially read from 10 files, interpolated to 1/20°grid and then scattered to each processor or thread. All the processes are done at the master processor. In the revised code, we use 10 different processes to read, interpolate and scatter parallelly, hence reduce the time to about 1/10 of the original. We will explain this in the revised manuscript.

105

Line 263, What is "the core-process"?

**Response:** Thanks. The core-process here indicated the computation routines within one integration step, and it does not include the daily-mean and I/O. We further revised the expression in the manuscript.

110

Line 389, What are "The dynamic core and physic packages"?

**Response:** Thanks. The former, the dynamic core, is the code to solve the equations numerically. The latter, the physic packages, is the code to compute the contributions of physic processes to the change of the circulation. Usually, these two
115  parts are coded separately in the oceanic or atmospheric general circulation models. We further revised the expression in the manuscript.

Some minor corrections:
Line 20, Change "can still obtain an increasing," to "can still be increased to"
120  Line 60, Change "is supported by" to "provides support for"
Line 81, Change "preparing" to "in preparation"
Line 109, Change "totally" to "in total,"
Line 111, grammar
Line 112, Change "are suitable for" to "require" or "demand"
125  Line 154, Remove "including"
Line 164, Change "earlier" to "ago"
Line 196, Change "grids were united " to "grid points are grouped"
Line 217, "parallel" to "parallelize"
Line 223, "parallel" to "parallelization"
130  Line 245, "Variable" to "Floating point" or "Arithmetic"
Line 259, "updated" to "increased"

Line 269, Change "0.2-0.3 SYPD will require too much" to "at a speed of 0.2-0.3 SYPD it will take too long"

**Response:** Done. Many thanks for your help in improving the language of the manuscript. All the improper usages have been
135 corrected in the revised manuscript following your suggestions.

[revised manuscript text omitted]

Page 7: [1] Deleted    liu hailong         2021/4/8 AM11:23:00

Page 7: [1] Deleted    liu hailong         2021/4/8 AM11:23:00

Page 7: [1] Deleted    liu hailong         2021/4/8 AM11:23:00

Page 7: [1] Deleted    liu hailong         2021/4/8 AM11:23:00

Page 7: [1] Deleted    liu hailong         2021/4/8 AM11:23:00

Page 7: [1] Deleted    liu hailong         2021/4/8 AM11:23:00

Page 7: [1] Deleted    liu hailong         2021/4/8 AM11:23:00

Page 7: [1] Deleted    liu hailong         2021/4/8 AM11:23:00

Page 7: [1] Deleted    liu hailong         2021/4/8 AM11:23:00

Page 7: [1] Deleted    liu hailong         2021/4/8 AM11:23:00

Page 7: [1] Deleted    liu hailong    2021/4/8 AM11:23:00

Page 7: [1] Deleted    liu hailong    2021/4/8 AM11:23:00

Page 7: [1] Deleted    liu hailong    2021/4/8 AM11:23:00

Page 7: [1] Deleted    liu hailong    2021/4/8 AM11:23:00

Page 7: [1] Deleted    liu hailong    2021/4/8 AM11:23:00

Page 7: [1] Deleted    liu hailong    2021/4/8 AM11:23:00

Page 7: [1] Deleted    liu hailong    2021/4/8 AM11:23:00

Page 7: [1] Deleted    liu hailong    2021/4/8 AM11:23:00

Page 7: [2] Deleted    liu hailong                2021/4/8 AM11:23:00

Page 7: [2] Deleted    liu hailong                2021/4/8 AM11:23:00

Page 7: [2] Deleted    liu hailong                2021/4/8 AM11:23:00

Page 7: [2] Deleted    liu hailong                2021/4/8 AM11:23:00

Page 7: [2] Deleted    liu hailong                2021/4/8 AM11:23:00

Page 7: [2] Deleted    liu hailong                2021/4/8 AM11:23:00

Page 7: [2] Deleted    liu hailong                2021/4/8 AM11:23:00

Page 7: [2] Deleted    liu hailong                2021/4/8 AM11:23:00

Page 7: [2] Deleted    liu hailong                2021/4/8 AM11:23:00

Page 7: [2] Deleted    liu hailong    2021/4/8 AM11:23:00

Page 7: [2] Deleted    liu hailong    2021/4/8 AM11:23:00

Page 7: [2] Deleted    liu hailong    2021/4/8 AM11:23:00

Page 7: [2] Deleted    liu hailong    2021/4/8 AM11:23:00

Page 7: [2] Deleted    liu hailong    2021/4/8 AM11:23:00

Page 7: [3] Deleted    liu hailong    2021/4/8 AM11:23:00

Page 7: [3] Deleted    liu hailong    2021/4/8 AM11:23:00

Page 7: [3] Deleted    liu hailong    2021/4/8 AM11:23:00

Page 7: [3] Deleted    liu hailong    2021/4/8 AM11:23:00

Page 7: [3] Deleted    liu hailong    2021/4/8 AM11:23:00

Page 7: [3] Deleted    liu hailong             2021/4/8 AM11:23:00

Page 7: [3] Deleted    liu hailong             2021/4/8 AM11:23:00

Page 7: [3] Deleted    liu hailong             2021/4/8 AM11:23:00

Page 7: [3] Deleted    liu hailong             2021/4/8 AM11:23:00

Page 7: [3] Deleted    liu hailong             2021/4/8 AM11:23:00

Page 7: [3] Deleted    liu hailong             2021/4/8 AM11:23:00

Page 7: [3] Deleted    liu hailong             2021/4/8 AM11:23:00

Page 7: [3] Deleted    liu hailong             2021/4/8 AM11:23:00

Page 7: [3] Deleted    liu hailong             2021/4/8 AM11:23:00

Page 7: [3] Deleted     liu hailong            2021/4/8 AM11:23:00

Page 14: [4] Formatted            liu hailong            2021/4/8 AM11:23:00

Font color: Text 1

Page 14: [5] Formatted            liu hailong            2021/4/8 AM11:23:00

Font color: Text 1

Page 14: [6] Formatted            liu hailong            2021/4/8 AM11:23:00

Font color: Text 1

Page 14: [6] Formatted            liu hailong            2021/4/8 AM11:23:00

Font color: Text 1

Page 14: [6] Formatted            liu hailong            2021/4/8 AM11:23:00

Font color: Text 1

Page 14: [7] Formatted            liu hailong            2021/4/8 AM11:23:00

Font color: Black

Page 14: [7] Formatted            liu hailong            2021/4/8 AM11:23:00

Font color: Black

Page 14: [8] Formatted            liu hailong            2021/4/8 AM11:23:00

| Page 14: [9] Formatted | liu hailong | 2021/4/8 AM11:23:00 |

Font color: Text 1

| Page 14: [10] Formatted | liu hailong | 2021/4/8 AM11:23:00 |

Font color: Text 1

| Page 14: [11] Formatted | liu hailong | 2021/4/8 AM11:23:00 |

Font color: Text 1

| Page 14: [12] Formatted | liu hailong | 2021/4/8 AM11:23:00 |

Font color: Text 1

| Page 14: [13] Formatted | liu hailong | 2021/4/8 AM11:23:00 |

Font color: Text 1

| Page 14: [13] Formatted | liu hailong | 2021/4/8 AM11:23:00 |

Font color: Text 1

| Page 14: [14] Formatted | liu hailong | 2021/4/8 AM11:23:00 |

Font color: Text 1

| Page 14: [15] Formatted | liu hailong | 2021/4/8 AM11:23:00 |

Font color: Text 1

| Page 14: [16] Formatted | liu hailong | 2021/4/8 AM11:23:00 |

Font color: Text 1

| Page 14: [16] Formatted | liu hailong | 2021/4/8 AM11:23:00 |

| Page 14: [16] Formatted | liu hailong | 2021/4/8 AM11:23:00 |

Font color: Text 1

| Page 14: [17] Formatted | liu hailong | 2021/4/8 AM11:23:00 |

Font color: Text 1

| Page 14: [18] Formatted | liu hailong | 2021/4/8 AM11:23:00 |

Font color: Text 1

| Page 14: [19] Formatted | liu hailong | 2021/4/8 AM11:23:00 |

Font color: Text 1

| Page 14: [20] Formatted | liu hailong | 2021/4/8 AM11:23:00 |

Font color: Text 1

| Page 14: [21] Deleted | liu hailong | 2021/4/8 AM11:23:00 |

| Page 14: [21] Deleted | liu hailong | 2021/4/8 AM11:23:00 |

| Page 14: [21] Deleted | liu hailong | 2021/4/8 AM11:23:00 |

| Page 14: [21] Deleted | liu hailong | 2021/4/8 AM11:23:00 |

| Page 14: [21] Deleted | liu hailong | 2021/4/8 AM11:23:00 |

| Page 14: [21] Deleted | liu hailong | 2021/4/8 AM11:23:00 |

| Page 14: [21] Deleted | liu hailong | 2021/4/8 AM11:23:00 |

| Page 14: [22] Formatted | liu hailong | 2021/4/8 AM11:23:00 |

Font color: Text 1

| Page 14: [22] Formatted | liu hailong | 2021/4/8 AM11:23:00 |

Font color: Text 1

| Page 14: [23] Formatted | liu hailong | 2021/4/8 AM11:23:00 |

Font color: Text 1

| Page 14: [23] Formatted | liu hailong | 2021/4/8 AM11:23:00 |

Font color: Text 1

| Page 14: [24] Formatted | liu hailong | 2021/4/8 AM11:23:00 |

Font color: Text 1

| Page 14: [25] Formatted | liu hailong | 2021/4/8 AM11:23:00 |

Font color: Text 1

| Page 14: [25] Formatted | liu hailong | 2021/4/8 AM11:23:00 |

Font color: Text 1

| Page 14: [26] Deleted | liu hailong | 2021/4/8 AM11:23:00 |

| Page 14: [26] Deleted | liu hailong | 2021/4/8 AM11:23:00 |

| Page 14: [26] Deleted | liu hailong | 2021/4/8 AM11:23:00 |

| Page 14: [26] Deleted | liu hailong | 2021/4/8 AM11:23:00 |

| Page 14: [27] Deleted | liu hailong | 2021/4/8 AM11:23:00 |

| Page 14: [27] Deleted | liu hailong | 2021/4/8 AM11:23:00 |

| Page 14: [27] Deleted | liu hailong | 2021/4/8 AM11:23:00 |

| Page 14: [28] Formatted | liu hailong | 2021/4/8 AM11:23:00 |

Font color: Text 1

| Page 14: [28] Formatted | liu hailong | 2021/4/8 AM11:23:00 |

| Page 15: [29] Formatted | liu hailong | 2021/4/8 AM11:23:00 |

Font color: Text 1

| Page 15: [30] Formatted | liu hailong | 2021/4/8 AM11:23:00 |

Font color: Text 1

| Page 15: [31] Deleted | liu hailong | 2021/4/8 AM11:23:00 |

| Page 15: [31] Deleted | liu hailong | 2021/4/8 AM11:23:00 |

| Page 15: [31] Deleted | liu hailong | 2021/4/8 AM11:23:00 |

| Page 15: [32] Deleted | liu hailong | 2021/4/8 AM11:23:00 |

| Page 15: [32] Deleted | liu hailong | 2021/4/8 AM11:23:00 |

| Page 15: [32] Deleted | liu hailong | 2021/4/8 AM11:23:00 |

| Page 15: [32] Deleted | liu hailong | 2021/4/8 AM11:23:00 |

| Page 15: [32] Deleted | liu hailong | 2021/4/8 AM11:23:00 |

| Page 15: [32] Deleted | liu hailong | 2021/4/8 AM11:23:00 |

| Page 15: [33] Formatted | liu hailong | 2021/4/8 AM11:23:00 |

Font color: Text 1

| Page 15: [33] Formatted | liu hailong | 2021/4/8 AM11:23:00 |

Font color: Text 1

| Page 15: [34] Formatted | liu hailong | 2021/4/8 AM11:23:00 |

Font color: Text 1

| Page 15: [35] Formatted | liu hailong | 2021/4/8 AM11:23:00 |

Font color: Text 1

| Page 15: [36] Formatted | liu hailong | 2021/4/8 AM11:23:00 |

Font color: Text 1

| Page 15: [37] Formatted | liu hailong | 2021/4/8 AM11:23:00 |

Font color: Text 1

| Page 15: [38] Formatted | liu hailong | 2021/4/8 AM11:23:00 |

Font color: Text 1

| Page 15: [39] Formatted | liu hailong | 2021/4/8 AM11:23:00 |
|---|---|---|

Font color: Text 1

| Page 15: [39] Formatted | liu hailong | 2021/4/8 AM11:23:00 |
|---|---|---|

Font color: Text 1

| Page 15: [40] Formatted | liu hailong | 2021/4/8 AM11:23:00 |
|---|---|---|

Font color: Text 1

| Page 15: [40] Formatted | liu hailong | 2021/4/8 AM11:23:00 |
|---|---|---|

Font color: Text 1

| Page 15: [41] Formatted | liu hailong | 2021/4/8 AM11:23:00 |
|---|---|---|

Font color: Text 1

| Page 15: [42] Formatted | liu hailong | 2021/4/8 AM11:23:00 |
|---|---|---|

Font color: Text 1

| Page 15: [43] Formatted | liu hailong | 2021/4/8 AM11:23:00 |
|---|---|---|

Font color: Text 1

| Page 15: [44] Formatted | liu hailong | 2021/4/8 AM11:23:00 |
|---|---|---|

Font color: Text 1

| Page 15: [45] Formatted | liu hailong | 2021/4/8 AM11:23:00 |
|---|---|---|

Font color: Text 1

| Page 15: [46] Formatted | liu hailong | 2021/4/8 AM11:23:00 |
|---|---|---|

| Page 15: [47] Formatted | liu hailong | 2021/4/8 AM11:23:00 |

Font color: Text 1

| Page 15: [47] Formatted | liu hailong | 2021/4/8 AM11:23:00 |

Font color: Text 1

| Page 15: [48] Formatted | liu hailong | 2021/4/8 AM11:23:00 |

Font color: Text 1

| Page 15: [49] Formatted | liu hailong | 2021/4/8 AM11:23:00 |

Font color: Text 1

| Page 15: [50] Formatted | liu hailong | 2021/4/8 AM11:23:00 |

Font color: Black

| Page 15: [50] Formatted | liu hailong | 2021/4/8 AM11:23:00 |

Font color: Black

| Page 15: [50] Formatted | liu hailong | 2021/4/8 AM11:23:00 |

Font color: Black

| Page 15: [50] Formatted | liu hailong | 2021/4/8 AM11:23:00 |

Font color: Black

| Page 15: [50] Formatted | liu hailong | 2021/4/8 AM11:23:00 |

Font color: Black

Page 15: [51] Deleted          liu hailong          2021/4/8 AM11:23:00

Page 15: [51] Deleted          liu hailong          2021/4/8 AM11:23:00

Page 15: [51] Deleted          liu hailong          2021/4/8 AM11:23:00

Page 15: [51] Deleted          liu hailong          2021/4/8 AM11:23:00

Page 15: [52] Deleted          liu hailong          2021/4/8 AM11:23:00

Page 15: [52] Deleted          liu hailong          2021/4/8 AM11:23:00